# Cross-Modal Knowledge Transfer for Scalable Text-Driven Multimodal Prompt Learning

## Abstract

Integrating prompt tuning with multimodal learning enhances generalization across downstream tasks. However, current approaches rely extensively on large-scale modality-specific labeled data (for instance, image, video, or audio) or specialize in single-modality adaptation. And, embeddings produced by encoders trained on different datasets with different architectures reside in incompatible, semantically orthogonal spaces. To address these, we propose a scalable framework, T2n-Modal to invigorate a universal representation model that supports unlimited modalities using only text data. Also, we establish theoretical guarantees with upper bounds on cross-modal transfer under incompatible spaces, and learnable bidirectional projection with orthogonal regularization. Our method integrates three key components, modality prompt pools, text construction mechanisms, and modality-aligned text encoders derived from pre-trained multimodal large models. This framework enables seamless extension to new modalities by augmenting prompt pools and corresponding text encoders. To ensure coherent learning across modalities, T2n-Modal employs intra- and inter-modal learning strategies, which preserve fine-grained category distinctions within modalities while enforcing semantic alignment between them. Leveraging its scalable architecture and pre-trained encoders, T2n-Modal efficiently generalizes to novel modalities without requiring labeled data. Specifically, despite using no modality-specific supervision, our method achieves state-of-the-art performance on diverse benchmarks including image, audio and video classification tasks.

## 1 Introduction

Recent advances in unimodal architectures Dosovitskiy et al. (2021); Vaswani et al. (2017); Gong et al. (2021); Arnab et al. (2021) and multimodal pre-training models Devlin et al. (2019); Radford et al. (2021); Tong et al. (2022) signifies strong representation capabilities in multimodal learning Li et al. (2023a); Zhou et al. (2022c); Jian et al. (2023); Zhu et al. (2023b; 2024); Elizalde et al. (2023); Wu et al. (2023). Prompt tuning Lester et al. (2021); Zhu et al. (2023a); Zhou et al. (2022a;b); Wang et al. (2023b) leverages aligned pretrained models Radford et al. (2021); Wu et al. (2023); Luo et al. (2020); Lin et al. (2021) in problems constrained by limited labeled data or computational resources to achieve robust generalization across downstream tasks—such as image Zhou et al. (2022b), audio Guo et al. (2023a), and video classification Huang et al. (2023) by optimizing only a minimal set of parameters. Despite its potential as an efficient paradigm for adapting large-scale pretrained models Radford et al. (2021); Dosovitskiy et al. (2021); Wu et al. (2023), current prompt-tuning methods remain dependent on extensive modality-specific labeled data (e.g., image, audio, or video). For instance, image-supervised methods Zhou et al. (2022b); Sun et al. (2022) construct text prompts combining them with textual labels to align pretrained models with labeled image data for classification tasks. Similarly, prior work Ju et al. (2022); Luo et al. (2020); Huang et al. (2023); Deshmukh et al. (2023); Yang et al. (2024a) adapts multimodal pretrained models to video and audio understanding tasks using supervised video and audio datasets. However, acquiring sufficient modality-specific labels requires significant manual effort, and scarcity of such labels can hinder the development of robust classification networks. Without any labeled data, these methods often fail entirely. Recent studies propose leveraging contrastive learning-based embedding spaces (e.g. CLIP) for prompt tuning Cherti et al. (2023) to miti-

gate this issue. For instance, TaI-DPT Guo et al. (2023b) introduces a method that trains text prompts using labeled textual data (e.g. COCO-Captions) rather than labeled images, enabling image-based testing with the learned prompts. To minimize reliance on manually annotated text, TOD Yang et al. (2024b) suggest employing synthetic text generated by large language models (LLMs) as a substitute. However, these approaches necessitate carefully constructed text or visual prompt frameworks along with dedicated text encoders for prompt embedding. Also, these existing methods are limited to single-modality tasks (e.g., image, audio, or video classification) requiring separate models for additional modalities.

We investigate a general representation model that scales seamlessly across unlimited modalities without requiring modality-specific labeled data. To achieve this, the model must satisfy three key conditions (a) The model must follow easily collected text data for training, completely evicting dependency on labeled datasets. (b) The model architecture maintains sufficient flexibility to integrate new modalities or categories while streamlining prompt construction, thereby minimizing prompt encoding complexity. (c) The model must prevent cross-modal interference during learning and employ tailored training strategies to enhance multimodal representation. Motivated by these considerations, we introduce Text to n-Modalities (T2n-Modal), a generalized representation learning framework capable of extending to arbitrarily many modalities using only text data synthesized by LLMs. In contrast to prior approaches such as TaI-DPT Guo et al. (2023b), which rely on laborious, multi-grained prompt engineering, our method simplifies modality integration by representing each category as a randomly initialized latent vector. By exploiting the instruction-following capabilities of modern LLMs Touvron et al. (2023), we efficiently generate textual training data for any target modality. Our framework optimizes these vectors directly within the aligned embedding space of pre-trained models Radford et al. (2021); Wu et al. (2023); Wang et al. (2023a), bypassing the need for intermediate encoding stages. Crucially, because all modality categories share an identical initialization protocol, T2n-Modal supports dynamic expansion to novel modalities without retraining existing category-specific prompts. To further enhance representation learning, we propose a unidirectional contrastive loss mechanism, wherein modalities with stronger inherent representational power guide the optimization of weaker ones. This approach not only improves the expressiveness of weaker modalities but also yields complementary gains in the performance of stronger modalities demonstrating a synergistic effect. We present a comprehensive empirical evaluation across multiple modalities, including image, audio, and video classification tasks. In a fully unsupervised setting, T2n-Modal demonstrates superior performance compared to both pretrained baselines and recent state-of-the-art methods Radford et al. (2021). For image classification, T2n-Modal achieves significant improvements over CLIP with gains of 12.8% on MS-COCO, 12.6% on NUS-WIDE and 9.0-12.0% on Objects365 datasets. In video classification, our model attains 1.5-3.0% higher Top-1 accuracy than ViCLIP on the Kinetics benchmarks. Similarly, for audio classification, T2n-Modal surpasses the pretrained CLAP model on ESC-50 Piczak (2015) and UrbanSound8K datasets. Also, our proposed framework exhibits seamless compatibility with supervised models enabling integration to enhance their classification performance through auxiliary unlabeled data. In summary, our contributions are as follows,

- We propose a novel framework, T2n-Modal to eliminate modality-specific labeled data entirely by employing LLM-generated text corpora through independent prompt pools, unidirectional contrastive learning from stronger to weaker modalities, and a ranking loss minimising computational overhead by 50%.

- We characterize the embedding space incompatibility problem via cross-modal alignment error when encoders are trained on disjoint semantic manifolds. We introduce learnable bidirectional projection layers with orthogonal regularization to certify closed-form optimal projections and guarantees of near-orthogonality.

- We provide theoretical guarantees for cross-modal transfer underlaying that inter-modal contrastive learning without projections collapses to implicit regularization.

## 2 Related Work

**Prompt Tuning:** Prompt tuning Zhou et al. (2022b); Duan et al. (2023) offers a parameter-efficient approach for adapting models to various downstream tasks. Recent advances demonstrate that learnable

context vectors can effectively align with image features using frozen CLIP encoders Radford et al. (2021); Zhou et al. (2022b). Under label-scarce conditions, TaI-DPT Guo et al. (2023b) achieve superior performance to pretrained MLMs in image and audio classification by employing multi-grained text prompts trained exclusively on text corpora. Wu et al. (2023) further advance image classification through joint optimization of pseudo-visual and text prompts. While these methods demonstrate promising results, they exhibit three key limitations, (a) dependence on large labeled datasets, (b) requirement for intricate prompt engineering, and (c) restriction to unimodal adaptation. Our framework addresses these constraints by (i) eliminating the prompt encoder entirely, (ii) enabling seamless multimodal scaling, and (iii) initializing all modality categories with a unified vector representation. Essentially, our approach operates solely on synthetic text data generated by LLMs bypassing the need for manually annotated training samples.

**Multimodal Classification:** Image classification seeks to identify all categorical elements within an image. Recent advances model label correlations by explicitly incorporating semantic dependencies through object proposals, semantic graphs, and transformer-driven architectures Tran et al. (2018). Under data scarcity conditions, researchers have developed methods to address more challenging learning scenarios including zero-shot, few-shot, and partial-label classification Simon et al. (2022). Also, DualCoOp Sun et al. (2022) and DualCoOp++ Hu et al. (2023) demonstrate performance gains in these settings by learning multiple discriminative prompts per class, effectively handling both zero-shot and partial-label classification tasks. Audio classification categorizes audio signals into distinct classes. Conventional approaches predominantly employ machine learning techniques with handcrafted feature extraction Henaff et al. (2011). Recent advances in deep learning have enabled researchers to investigate neural network-based methods for this task Chen et al. (2022); Hanif et al. (2024). Furthermore, several studies now apply transformer architectures to audio classification, effectively modeling long-range temporal dependencies in audio sequences Gong et al. (2021); Elizalde et al. (2023); Duan et al. (2023). Video classification requires recognizing actions within video sequences. Early approaches primarily developed two-stream networks and 3D CNNs for action recognition. Yan et al. (2022) Following the success of vision transformers in image tasks Liu et al. (2022), recent work has investigated effective adaptation strategies to transfer pre-trained image models to video understanding. To address local redundancy in videos, UniFormerV2 proposes local and global relation aggregators that learn discriminative spatiotemporal representations Li et al. (2022).

## 3 Our Method

### 3.1 Preliminaries

Let $\mathcal{X}_I$, $\mathcal{X}_A$, $\mathcal{X}_V$ denote the input spaces for image, audio, and video modalities respectively. For each modality $m \in \{I, A, V\}$, we have a pretrained frozen encoder $\mathcal{E}_m : \mathcal{X}_m \to \mathbb{R}^d$ that maps inputs to a $d$-dimensional embedding space. These encoders are trained independently on different datasets $\mathcal{D}_m$ with different architectures.

**Definition 1** (Embedding Space Incompatibility). *Let $\mathcal{E}_I$, $\mathcal{E}_A$, $\mathcal{E}_V$ be three encoders with output space $\mathbb{R}^d$. We define the pairwise semantic incompatibility as,*

$$\delta_{mn} = \inf_{P \in \mathcal{P}_{d \times d}} \mathbb{E}_{x \sim \mathcal{X}_m, y \sim \mathcal{X}_n} \|P\mathcal{E}_m(x) - \mathcal{E}_n(y)\|_2^2,$$

*where $\mathcal{P}_{d \times d}$ denotes the set of orthogonal projection matrices. If $\delta_{mn} > 0$ for any pair $(m, n)$, the embedding spaces are semantically incompatible.*

Our claim of the analysis is that for independently trained encoders, $\delta_{mn}$ is bounded below by a positive constant determined by the alignment between training distributions.

**Theorem 1** (Orthogonality of Heterogeneous Embedding Spaces). *Let $\mathcal{E}_I$ and $\mathcal{E}_A$ be two encoders trained on datasets $\mathcal{D}_I$ and $\mathcal{D}_A$ with distinct distributions $P_I$ and $P_A$ supported on disjoint or weakly-overlapping semantic manifolds. Assume the encoders are Lipschitz with constants $L_I, L_A$ and that the datasets satisfy $supp(P_I) \cap supp(P_A) = \emptyset$ up to a set of measure $\epsilon$. Then there exists $\gamma > 0$ such that for any linear map $W : \mathbb{R}^d \to \mathbb{R}^d$,*

$$\mathbb{E}_{x \sim P_I, y \sim P_A} \|W\mathcal{E}_I(x) - \mathcal{E}_A(y)\|^2 \geq \gamma \cdot \min\{L_I, L_A\} \cdot d_{\mathcal{W}}(P_I, P_A) + \Omega(1),$$

*where $d_{\mathcal{W}}(\cdot, \cdot)$ denotes the Wasserstein distance.*

The construction of the Wasserstein lower bound and analysis of the disjoint support condition is detailed in appendix B.

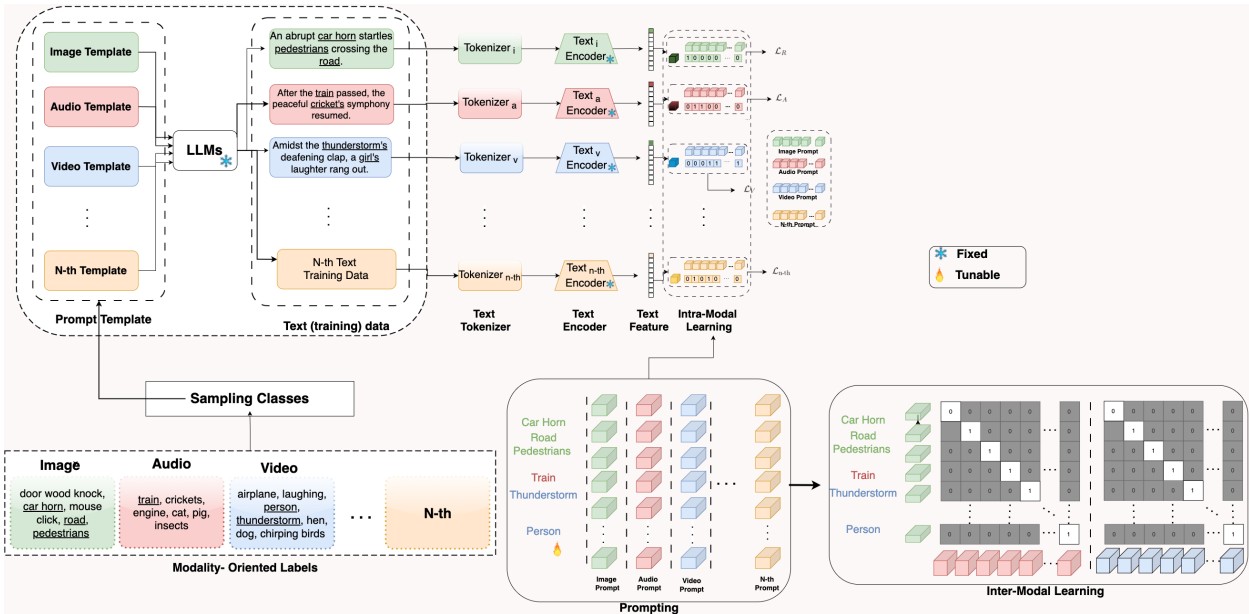

Figure 1: Overall framework of T2n-Modal framework which employs class-driven prompts to represent arbitrary categories with text data generated automatically via large language models. The intra-modal learning component optimizes each modality's prompt pool using features extracted from pretrained multimodal large models (MLMs). Simultaneously, inter-modal learning transfers knowledge from high-performance (stronger) modalities to improve the representation quality of underperforming (weaker) ones through cross-modal alignment.

## 3.2  Problem Settings

The T2n-Modal is a general representation learning framework designed to handle multiple modalities through text-guided prompt learning. Its architecture in Fig. 1 comprises three core components, (a) LLM-driven data construction, where large language models generate semantically rich training data, (b) initiating prompt and modality-aware text encoding which aligns textual descriptors with target modalities, and (c) inter- and intra-modal learning enabling the model to capture both domain-specific features and cross-modal relationships. Each prompt uniquely represents a distinct class and so we learn the modality-oriented prompt pools. To elucidate this framework, we present the video classification pipeline as a representative case. For a given input video, we first extract its visual features using the ViCLIP video encoder Wang et al. (2023a) (substituting the modality-oriented text encoder (Fig. 1). We then compute the cosine similarity between these video features and all prompts in the video prompt pool, assigning the class corresponding to the highest-scoring prompt as the prediction. Conclusively, this process requires no additional encoding steps for the prompts during inference as they directly interact with the video features. This architectural design yields significant improvements in computational efficiency during inference. We apply this identical framework to image and audio classification, where features from each modality are similarly matched against their respective prompt pools to generate predictions.

## 3.3  Training Data Curation via LLMs

We explore a method for generating high-quality text training data tailored to modality-specific class labels. Unlike prior approaches such as TaI-DPT Guo et al. (2023b), which rely on noun filtering. Our framework

employs structured prompt templates to guide large language models in producing semantically relevant sentences that explicitly incorporate target labels. We design a structured prompt template to guide LLaMA-2-7B Touvron et al. (2023) in generating textual descriptions for multimodal data. For a given label set, we design the following query template, Template: *Generate five concise English sentences (each $\leq$ 25 words) describing a {Modality}. Each sentence must include: {Labels}.* Here, {Modality} is instantiated with domain-specific terms (e.g., 'image', 'video', 'audio'), while {Labels} denotes modality-dependent class labels, constrained to a maximum of 3 labels for image/audio modalities and 2 labels for video. Our approach offers two key benefits over noun filtering, firstly, mitigation of lexical diversity issues (i.e., singular/plural forms) inherent to direct noun extraction. And, secondly, elimination of preprocessing overhead for complex phrases in video/audio descriptions. This ensures the model captures both inter-category dependencies (e.g., contextual relationships) and intra-category discriminative features. By leveraging LLM-generated sentences grounded in the template's Labels, we ensure that each sentence is intrinsically linked to its corresponding ground-truth labels enhancing both scalability and semantic coherence.

**Prompt Encoding:** We elucidate the extensibility of T2n-Modal to arbitrary modalities, we consider image (i), audio (a), and video (v) as representative examples. For each modality $m \in \{i, a, v\}$, a dedicated prompt pool $Q_m$ defined as

$$Q_m = \{q_1^m, q_2^m, q_3^m, \ldots, q_N^m\} \tag{1}$$

where each $q_i^m \in \mathbb{R}^d$ corresponds to a learnable $i$-th class-oriented text query (prompt) embedding and $N$ denotes the pool size (overall number of labels). The prompt pool maintains a fixed length across all modalities ($Q_m \in \mathbb{R}^{N \times d}$, where $m \in \{i, a, v\}$), confining labels from each modality. To accommodate a new modality, a dedicated prompt pool is initialized ensuring no interference with previously learned prompt pools. Similarly, the introduction of a new label triggers the addition of a class-oriented prompt to every existing pool preserving the integrity of prior class-oriented prompts. This design ensures T2n-Modal supports scalable extension to arbitrary modalities and categories without compromising stability. Prior approaches commonly treat text as a proxy representation for other modalities (images or audio) in zero-shot classification tasks. This paradigm implicitly assumes that pretrained models have effectively aligned textual and non-textual modalities within a shared embedding space, thereby enabling text-derived features to approximate those of other modalities. However, such methods are typically modality-oriented and do not leverage cross-modal complementary relationships. To address this limitation, we propose a parallel architecture that integrates modality-oriented text encoders ($\mathcal{T}_i, \mathcal{T}_a, \mathcal{T}_v$) derived from pretrained models, CLIP (image) Cherti et al. (2023), CLAP (audio) Wu et al. (2023), and ViCLIP (video) Wang et al. (2023a) to extract task-adaptive textual features. Our analysis reveals that CLIP and CLAP exhibit superior representational capabilities for image and audio modalities compared to ViCLIP for video as evidenced by their zero-shot classification performance. Building upon this observation, we propose a unidirectional learning strategy, wherein representations from stronger modalities guide the learning process of weaker ones. Our results demonstrate that this approach enhances performance across all modalities simultaneously suggesting that cross-modal knowledge transfer can mitigate inherent disparities in unimodal representation quality.

### 3.4 Learning Strategy for Various Modalities

To learn the modality-oriented prompt pools, we propose a dual-objective learning framework, (a) inter-modal learning to enhance the representational robustness of weaker modalities through knowledge transfer from semantically richer modalities, and (b) intra-modal learning to optimize prompt pools for individual modalities by aligning them with global text features encoded by modality-oriented text encoders.

**Intra-modal Ranking and Learning:** We adopt the image modality as a representative case to elucidate the intra-modal learning framework, provided that the same methodology extends to video and audio modalities. Let $\mathcal{P} = \{p_1, p_2, \ldots, p_N\}$ denote the unified label space, where $N$ represents the overall number of labels encompassing all modalities. The textual training corpus for image labels is formalized as $\mathbb{L} = \{l_i, y_i\}_{i=1}^T$, where $T$ denotes the number of text instances. Here, $y_i = \{y_{i_1}, y_{i_2}, y_{i_3}, \ldots, y_{i_N}\}$ represents the ground-truth multi-hot label vector for text $l_i$ with $y_{ij} = 1$ if $l_i$ is generated from label $p_j$ (and 0 in other cases). The textual embedding for each $l_i$ is obtained via the frozen CLIP text encoder expressed as $g_i = \mathcal{T}_\phi(l_i)$. Let $\mathcal{T}_\phi(\cdot)$ denote the pretrained CLIP text encoder and $g_i \in \mathbb{R}^d$ represent the normalized global text feature corresponding to $l_i$, where $d$ is the feature dimension. For audio or video modality inputs, we substitute

$\mathcal{T}_\phi(\cdot)$ with text encoders from ViCLIP or CLAP to extract modality-oriented text features. The similarity between $l_i$ and the image modality prompt pool is subsequently estimated as,

$$\text{sim}_{ij} = \langle g_i, p_j \rangle \ \forall j \in \{1, 2, \ldots, N\} \tag{2}$$

Let $p_j$ denote the $j-$th prompt in the image modality's prompt pool. Given these prompts can be optimized directly without requiring processing through an encoder or multilayer perceptron (MLP). In contrast to prior works Guo et al. (2023a); Yang et al. (2024a), which rely on intricate multi-grained prompt designs and computationally expensive encoding procedures. Our approach simplifies prompt architecture and reduces computational overhead by approximately 50%. For prompt optimization, we adopt a ranking loss in lieu of cross-entropy loss or InfoNCE. This selection is motivated by two key shortcomings of existing methods, where cross-entropy loss exclusively optimizes positive labels discarding negative label contributions which impedes convergence speed. For InfoNCE loss, it necessitates large-scale negative sampling and high-cost Softmax computations for effective optimization. By employing ranking loss, we directly optimize the relative similarity between positive and negative pairs, thereby improving both efficiency and convergence dynamics.

$$\mathcal{L}_R = \frac{1}{M} \sum_{k=1}^{M} \sum_{i \in b^+} \sum_{j \in b^-} \max(0, m - \text{sim}_{ki} + \text{sim}_{kj}) \tag{3}$$

Let $b^+$ denote the set of positive labels, where $y_{ij}$ for $j \in \{1, 2, \ldots, N\}$ and $b^-$ represent the set of negative labels. Here, $\text{sim}_{ki}$ and $\text{sim}_{kj}$ correspond to the positive and negative pair similarities as defined in Eq. 2, while $m$ denotes the margin parameter that quantifies the discrepancy between similarity pairs. For the audio and video modalities, we replace the text encoder $\mathcal{T}_\phi$ from Eq. 2 with the text encoders of CLAP and ViCLIP to extract text features. These features are then used to compute their respective similarities with the audio and video prompt pools. Consequently, we derive the ranking loss - $\mathcal{L}_R, \mathcal{L}_A, \mathcal{L}_V$ delineating an image, audio and video modalities. The total intra-modal learning loss is thus formulated as follows,

$$\mathcal{L}_{\text{intra-model}} = \mathcal{L}_R + \mathcal{L}_A + \mathcal{L}_V \tag{4}$$

In the training phase, we freeze the text encoders and optimize the modality-oriented prompt pools via the objective function in Eq. 4. The positive labels (Eq. 3) consist solely of image category labels, whereas the negative labels incorporate both negative image samples and cross-modal samples from other modalities. Employing other modalities' labels as negative examples serves to both increase the negative pair sample size and improve the video modality's representational capacity through cross-modal contrast. We apply this same principle to the audio and image modalities demonstrating its generalizability across multiple modalities.

**Inter-modal Unidirectional Contrastive Learning:** Contrastive learning aligns multimodal representations (for instance, image-text, video-text, and audio-text pairs) within a common embedding space. However, inherent disparities in information content across visual, auditory, and temporal modalities create a pronounced modality gap, particularly between video and text representations which degrades zero-shot classification performance. To address this limitation, we introduce unidirectional contrastive learning, a framework that transfers representational knowledge from stronger to weaker modalities. Our method dynamically identifies the weakest modality during training by monitoring validation metrics, where video consistently emerges as the underperforming modality compared to image and audio. For distinctness, we reformulate Eq 1 as,

$$
\begin{aligned}
Q_I &= \{q_{v_1}^I, q_{v_2}^I, \ldots, q_{v_v}^I, q_{a_1}^I, q_{a_2}^I, \ldots, q_{a_a}^I, q_{w_1}^I, q_{w_2}^I, \ldots, q_{w_w}^I\}, \\
Q_A &= \{q_{v_1}^A, q_{v_2}^A, \ldots, q_{v_v}^A, q_{a_1}^A, q_{a_2}^A, \ldots, q_{a_a}^A, q_{w_1}^A, q_{w_2}^A, \ldots, q_{w_w}^A\}, \\
Q_V &= \{q_{v_1}^V, q_{v_2}^V, \ldots, q_{v_v}^V, q_{a_1}^V, q_{a_2}^V, \ldots, q_{a_a}^V, q_{w_1}^V, q_{w_2}^V, \ldots, q_{w_w}^V\}
\end{aligned}
\tag{5}
$$

where $a + v + w = M$, and $q_k^I, q_k^A, q_k^V$ denote the class-oriented prompts from the image, audio and video prompt pools respectively. We initialize identical prompt pools across all modalities given each pool contains v video labels, a audio labels, and w image labels. Consequently, the video, audio, and image modalities share

the same composite prompt structure during initialization. We formulate a unidirectional contrastive learning objective using $Q_V$ and $Q_A$ (the query matrices). The similarity matrix is computed as $Q_A^T Q_V \in \mathbb{R}^{N \times N}$, where $N$ denotes the overall number of class labels. The ground truth alignment for $Q_V$ and $Q_A$ corresponds to an $N \times N$ diagonal matrix. Crucially, both the similarity matrix and ground truth matrix are batch-size invariant, their dimensions depend solely on the label cardinality N. For each video prompt in $Q_V$ and audio prompt in $Q_A$. we define the softmax normalized video-to-audio similarity distribution and ground truth alignment matrix as,

$$q_{ij}^{v \to a} = \frac{\exp(\text{sim}(v_i, a_j)/\theta)}{\text{sum}_{k=1}^{N} \exp(\text{sim}(v_i, a_k)/\theta)} \qquad y_{ij}^{v \to a} = \begin{pmatrix} 0 & 0 & \dots & 0 & \dots & 0 \\ 0 & \ddots & \ddots & \vdots & & \vdots \\ \vdots & \ddots & 0 & 0 & \dots & 0 \\ 0 & \dots & 0 & 1 & \dots & 0 \\ \vdots & & \vdots & \vdots & \ddots & \vdots \\ 0 & \dots & 0 & 0 & \dots & 1 \end{pmatrix}$$

where $a_j$ and $v_i$ represent the audio and video prompt embeddings, $\text{sim}(.,.)$ denotes the similarity function, and $\theta$ is a learnable temperature parameter. Unlike conventional contrastive learning frameworks that use an identity matrix for label supervision, our ground-truth label matrix $y_{ij}^{v \to a}$ explicitly sets the first $v + w$ diagonal elements to zero. This modification ensures that the loss computation disregards these positions during cross-entropy optimization. We define the unidirectional contrastive loss between video prompt pool $Q_V$ and audio prompt pool $Q_A$ as $\mathcal{L}_{v \to a} = \mathcal{L}_{\text{CE}}(y^{v \to a}, q^{v \to a})$, where $y_{ij}^{v \to a} \in \{0, 1\} \ \forall i, j \in \{1, 2, \dots, N\}$ denotes the ground-truth similarity score between video prompt $v_i$ and audio prompt $a_j$. Similarly, we derive the unidirectional contrastive loss between video prompt pool $Q_V$ and image prompt pool $Q_I$ as $\mathcal{L}_{v \to w} = \mathcal{L}_{\text{CE}}(y^{v \to w}, q^{v \to w})$. The total inter-modal learning loss combines these objectives as,

$$\mathcal{L}_{\text{inter-model}} = \mathcal{L}_{v \to a} + \mathcal{L}_{v \to w} \tag{6}$$

We explicitly align the image-label prompts in the video prompt pool with their counterparts in the image prompt pool, and similarly align the audio-label prompts with those in the audio prompt pool. These aligned cross-modal prompts serve as negative samples during video prompt pool optimization in the intra-modal learning phase, effectively increasing the negative pair count. To preserve the integrity of video-label prompt learning, we zero the diagonal elements corresponding to video-image pairs in the ground truth matrix. Our training protocol employs unidirectional contrastive learning for both video-to-audio and video-to-image relations. The complete objective function combines both learning paradigms,

$$\mathcal{L}_{\text{entire}} = \lambda_1 \mathcal{L}_{\text{intra-model}} + \lambda_2 \mathcal{L}_{\text{inter-model}} \tag{7}$$

where $\lambda_1$ and $\lambda_2$ weight the intra-modal and inter-modal loss components respectively.

We propose a theoretically-grounded version of learnable projection layers $P_{m \to n} : \mathbb{R}^d \to \mathbb{R}^d$ that map between embedding spaces trained with orthogonal regularization.

**Definition 2** (Projected Prompt Alignment). *For each ordered pair of modalities $(m, n)$, define a learnable linear projection $W_{m \to n} \in \mathbb{R}^{d \times d}$ constrained to be near-orthogonal*

$$\mathcal{R}_{orth}(W) = \|WW^\top - I\|_F^2$$

*The inter-modal loss becomes* $\mathcal{L}_{inter}^{proj} = \mathcal{L}_{CE}(y^{v \to a}, \tilde{q}^{v \to a}) + \mathcal{L}_{CE}(y^{v \to w}, \tilde{q}^{v \to w})$, *where* $\tilde{q}_{ij}^{v \to a} = \frac{\exp(sim(W_{v \to a} v_i, a_j)/\theta)}{\sum_k \exp(sim(W_{v \to a} v_i, a_k)/\theta)}$.

**Theorem 2** (Optimal Projection). *For any pair of encoders $\mathcal{E}_m, \mathcal{E}_n$ with joint distribution $P_{mn}$ over paired data (which can be empty), there exists a linear projection $W_{m \to n}^*$ that minimizes,*

$$\mathcal{J}(W) = \mathbb{E}_{(x_m, x_n) \sim P_{mn}} \|W \mathcal{E}_m(x_m) - \mathcal{E}_n(x_n)\|^2 + \beta \|WW^\top - I\|_F^2$$

*Also, $W^*$ satisfies $W^* = \Sigma_{mn} \Sigma_m^{-1} + \mathcal{O}(\beta)$ where $\Sigma_{mn} = \mathbb{E}[\mathcal{E}_n(x_n) \mathcal{E}_m(x_m)^\top]$ and $\Sigma_m = \mathbb{E}[\mathcal{E}_m(x_m) \mathcal{E}_m(x_m)^\top]$.*

We solve the quadratic optimization via matrix calculus and the proof is deferred to appendix B.

We showcase the intra-/inter-modal learning process in Fig. 2. For inter-modal learning, we implement unidirectional contrastive learning to align negative image/audio labels from the video prompt pool with both positive and negative labels from the image and audio prompt pools. This process effectively transfers representational knowledge from the image and audio prompt pools to the video prompt pool. For intra-modal learning (for instance, using the image modality), we incorporate negative image/video/audio labels in the image prompt pool. While the negative video and audio labels do not require explicit modality alignment, they serve to expand the negative sample space thereby improving the robustness of positive image label representations. Similarly, for the video modality, negative labels consist of aligned negative image/audio labels alongside native negative video labels. Essentially, this unidirectional contrastive learning strategy prevents negative image/audio labels in the video prompt pool from interfering with the learning of positive labels in the image/audio prompt pools.

## 4 Experiments

**Datasets:** We evaluate our method through extensive experiments across ten benchmark datasets spanning video, image, and audio modalities. For video classification, we employ both UCF101 Soomro et al. (2012) and the large-scale Kinetics-400 dataset Carreira & Zisserman (2017). Our image classification benchmarks include not only MS-COCO, VOC2007, and NUS-WIDE Guo et al. (2023a); Wu et al. (2023) but also VOC2012, mini-ImageNet Russakovsky et al. (2015), and Objects365 to ensure comprehensive evaluation. For audio classification, we utilize the ESC-50 and UrbanSound8K datasets. In all cases, we perform evaluation on publicly available test sets, where test labels are unavailable, we instead report performance on the validation sets. For images, we employ MS-COCO a comprehensive dataset encompassing 328k images, 2.5 million instances, and 80 object classes partitioned into 82,783 training images, 40,504 validation images, and 40,775 test images. We evaluation it using the validation set. The PASCAL VOC 2007 dataset comprising 9,963 images across 20 classes (5,011 for training and 4,952 for testing) is used for evaluation on its test set. Similarly, PASCAL VOC 2012 provides 11,540 images (5,717 training, 5,823 testing) across 20 categories with evaluation performed on the test set. NUS-WIDE contains 269,648 Flickr images annotated with 81 concepts divided into 161,789 training and 107,859 validation images. The validation set is used for assessment. mini-ImageNet, a subset of ImageNet features 60k images distributed across 100 classes (600 samples per class), typically with an 8:2 training-to-validation split and is evaluated on its test set. Finally, Objects365, a large-scale object detection dataset which includes 638k images, 365 object classes, and 10.1 million bounding boxes with 600k images for training, 38k for validation, and 100k for testing. Its evaluation is performed on the test set. For audio datasets, we employ the ESC-50 and UrbanSound8k datasets were utilized for environmental sound classification. ESC-50 comprises 2k audio samples uniformly distributed across 50 distinct environmental categories with each sample lasting up to 5 seconds. This dataset encompasses a broad spectrum of sounds including animal vocalizations, traffic noise, and indoor activities. For model evaluation, the validation set of ESC-50 was employed. UrbanSound8k, an open source dataset features 8,732 audio clips, each approximately 4 seconds in duration, categorized into ten urban sound classes such as air conditioning and car horns. The test set of UrbanSound8k was used for evaluating the developed methodologies. For videos, we employ UCF101 and Kinetics-400 datasets. UCF101 comprises 13,320 video clips categorized into 101 distinct action classes, with each class containing approximately 100 to 300 clips. These real-world video clips sourced from YouTube typically range from 10 to 30 seconds in duration, and all available data are used for evaluation. Kinetics-400 is a larger high-quality dataset also compiled from YouTube featuring 400 human action classes. This dataset offers a broader spectrum of actions, such as playing instruments, human-object interactions, and handshakes. Each action class within Kinetics-400 con-



Figure 2: A pictorial representation of the overall learning process

tains between 450 and 1150 video clips, segmented into training (250–1000 clips), validation (50 clips), and testing (100 clips) sets. The validation set of Kinetics-400 is specifically employed for method evaluation.

**LLM-driven Text Data Generation:** We train T2n-Modal using text data generated by LLMs rather than modality-oriented labeled data. To evaluate the impact of text corpus size, we conduct experiments on Kinetics 400, MS-COCO, and ESC-50 with varying amounts of training text. On Kinetics 400, a small text corpus (1K samples) yields only 9.8% Top-1 accuracy as the limited text per class fails to capture robust class-driven representations. However, increasing the text corpus size progressively improves performance at 100K samples, T2n-Modal surpasses zero-shot ViCLIP. For MS-COCO (80 classes) and ESC-50 (50 classes), a text corpus of 5K samples already exceeds zero-shot CLIP and CLAP by 7% mAP and 2% Top-1 accuracy, respectively. Performance plateaus at 50K samples suggesting that datasets with more classes require larger text corpora for optimal representation learning.

## 4.1 Experimental Settings

We employ the LAION released pretrained MLMs Schuhmann et al. (2022) as our modality-oriented encoders ViCLIP-Base Wang et al. (2023b) for video processing, CLIP-ViT-B-32 Cherti et al. (2023) for image analysis, and CLAP Wu et al. (2023) for audio understanding. For text generation, we utilize LLaMA-2-7B to produce 100k modality-oriented sentences requiring approximately 100 minutes of computation on a single NVIDIA A6000 GPU. By incorporating spatial relationship instructions into the generation templates, LLaMA-2-7B effectively produces text descriptions with accurate spatial representations. We initialize each class-driven prompt as a 512-dimensional vector sampled from a normal distribution ($\mu$=0, $\sigma$=0.02). Throughout training, we freeze all modality-aligned text encoders and optimize only the prompt parameters. CLIP BASELINE: We employ the publicly available CLIP model Radford et al. (2021) from LAION as our baseline which consists of a dual-encoder architecture, a Vision Transformer (ViT) image encoder and a transformer-driven text encoder. Both encoders contain 12 attention layers with a hidden dimension of 512. The image encoder processes input images resized to 224×224 pixels with a patch size of 32. For our experiments, we specifically adopt the CLIP-ViT-B-32 variant as the image encoder and paired text encoder. For audio-language pretraining, we adopt Contrastive Language-Audio Pretraining (CLAP) from the LAION as our baseline model. The architecture employs a transformer-driven audio encoder comprising four swin transformer block groups paired with a RoBERTa text encoder. We project both audio and text outputs into a shared 512-dimensional embedding space using two-layer MLPs with ReLU activation. For audio modality processing, we utilize CLAP audio encoder of LAION while retaining its native RoBERTa model for text encoding. For video modality, we leverage ViCLIP that extends the OpenAI CLIP architecture for video-language pretraining. The model incorporates a video encoder paired with a text encoder pre-trained on the InternVid dataset, a collection of 7 million videos annotated with detailed textual descriptions. For our baseline implementation, we adopt the BASE architecture configuration, which comprises 12 attention layers and 512-dimensional embeddings.

## 4.2 Results

We evaluate the zero-shot performance of T2n-Modal against pretrained multimodal baselines (ViCLIP, CLIP, CLAP), and compare its efficacy with state-of-the-art methods on image and audio classification tasks. Although, no prior work has investigated a comparable training paradigm, specifically prompt tuning exclusively with text data. Consequently, we adopt ViCLIP as the sole zero-shot benchmark for video-related comparisons as it represents the most relevant baseline under this constraint. We describe on the number of labels per modality in our experiments. Given that our results (Table 2) demonstrate that optimal performance is achieved with 2 labels for video and 3 labels for image/audio queries, as shown below (Top-1/Top-5 accuracy). This design strategy stems from modality-driven characteristics, for video, complex actions ("playing guitar") require concise labels to avoid semantic dilution. For image/audio, multiple object categories often coexist ("dog" + "park") justifying higher label counts to capture inter-category dependencies. Reducing labels degrades performance particularly for video, highlighting the trade-off between label quantity and semantic precision.

**Zero-shot Image Classification:** We present comparative results against state-of-the-art methods such as CLIP Radford et al. (2021), TaI-DPT Guo et al. (2023b), SPARC Miller et al. (2025) in Ta-

Table 1: We evaluate our proposed T2n-Modal against few-shot learning methods on standard benchmark datasets reporting mAP as the primary metric. Our method consistently achieves superior performance across all few-shot settings with the highest mAP scores highlighted in bold.

| Model
Dataset | Zero-shot CLIP | TaI-DPT | SPARC Miller et al. (2025) | T2n-Modal (Ours) |
|---|---|---|---|---|
| | Mean Average Precision (mAP) | | | |
| MS-COCO | 55.6 | 65.1 | 68.3 | 68.4 |
| VOC2007 | 80.5 | 88.3 | 89.2 | 89.5 |
| VOC2012 | 80.1 | 85.1 | - | 87.9 |
| NUS-WIDE | 37.1 | 46.5 | 47.2 | 49.7 |
| mini-ImageNet Russakovsky et al. (2015) | (85.5,94.3) | (86.2,94.7) | - | (90.5,98.4) |
| Objects365 | 19.8 | 24.1 | - | 28.3 |

Table 3: Comparison of T2n-Modal with Baselines for Zero-shot Audio Classification

| Model
Dataset | Zero-shot CLAP Elizalde et al. (2023) | PALM Hanif et al. (2024) | T2n-Modal (Ours) |
|---|---|---|---|
| | Mean Average Precision (mAP) | | |
| ESC-50 | 90.5 | 95.9 | 96.2 |
| UrbanSound8K | 76.2 | 80.8 | 85.4 |

ble 1. These baselines employ either complex prompt engineering or adapter modules, while Zero-shot CLIP uses the default prompt template "a photo of a [CLASS]". Our T2n-Modal significantly outperforms Zero-shot CLIP achieving gains of 12.8% and 12.6% mAP on MS-COCO and NUS-WIDE respectively. On VOC2007 and VOC2012 (20 classes), we observe consistent improvements of 8.0–10.0% over Zero-shot CLIP. For large-scale benchmarks, T2n-Modal maintains strong performance with 90.5% Top-1 accuracy on mini-ImageNet (vs. 85.5% for ZS-CLIP) and 28.6% mAP on Objects365 (vs. 19.8%). Also, while prior text-only training methods rely on intricate prompt design or encoding pipelines, our approach achieves state-of-the-art results across most datasets without such overhead.

**Zero-shot Audio Classification:** Table 3 presents the zero-shot audio classification results comparing CLAP and the recent state-of-the-art method PALM Hanif et al. (2024). Our proposed T2n-Modal achieves superior per-

Table 2: Performance across different configurations for video, image, and audio tasks.

| (Video, Image, Audio) | Kinetics 400 | MS-COCO | ESC-50 |
|---|---|---|---|
| (2, 3, 3) | (55.3, 80.5) | 68.2 | 94.4 |
| (1, 2, 2) | (43.8, 71.9) | 64.2 | 89.7 |
| (1, 1, 1) | (42.6, 69.5) | 59.6 | 87.3 |

formance exceeding Zero-shot CLAP Elizalde et al. (2023) by 4.1% on ESC-50 and 9.4% on UrbanSound8K despite CLAP's strong baseline accuracy. Also, T2n-Modal also outperforms PALM by 4.6% on UrbanSound8K dataset and 0.3% on ESC-50 without requiring complex prompt engineering.

**Zero-shot Video Classification:** We evaluate ViCLIP's zero-shot performance using the default prompt "a video of a [CLASS]". As shown in Table 9, our approach achieves consistent improvements over ZS-ViCLIP with +2.3% (Top-1) and +2.5% (Top-5) accuracy gains on UCF101. For the larger Kinetics-400 benchmark (400 classes), T2n-Modal further outperforms ZS-ViCLIP by 1.5%–3.0% in both Top-1 and Top-5 accuracy across all evaluated splits. These results demonstrate effectiveness of our approach in zero-shot video recognition without requiring labeled video data.

## 4.3 Analysis with Supervised Methods

Building on the methodology of TaI-DPT Guo et al. (2023b), we incorporate T2n-Modal with existing supervised models to enhance their performance. Consider a video with $n$ associated labels. We represent softmax predictions of the supervised model as $O_S = \{o_{s_1}, o_{s_2}, \ldots, o_{s_n}\}$. Also, for T2n-Modal, we compute the similarity between the video and n class-specific video prompts yielding softmax predictions $O_T = \{o_{t_1}, o_{t_2}, \ldots, o_{t_n}\}$. Subsequently, we compute the assimilated results $O_I$ by summing the corresponding softmax predictions, $O_I = \{o_{s_1} + o_{t_1}, o_{s_2} + o_{t_2}, \ldots, o_{s_n} + o_{t_n}\}$.

**Image Classification:** Table 4 presents our evaluation using DualCoOp++ Hu et al. (2023) which replaces

Table 4: We evaluate mAP under varying levels of label availability (10%-90%) across all benchmark datasets. The reported performance metrics for +TaI-DPT and +T2n-Modal incorporate predictions from both the respective models - TaI-DPT and DualCoOp++[†]. Our experimental results demonstrate that T2n-Modal consistently outperforms all state-of-the-art methods achieving superior performance on all major benchmarks.

| Model Dataset | | mAP with Partial available labels | | | | | | | | | |
|---|---|---|---|---|---|---|---|---|---|---|---|
| | | 10% | 20% | 30% | 40% | 50% | 60% | 70% | 80% | 90% | Avg. |
| MS-COCO | DualCoOp | 81.0 | 82.3 | 82.9 | 83.4 | 83.5 | 83.9 | 84.0 | 84.1 | 84.3 | 83.3 |
| | DualCoOp++ | 81.4 | 83.1 | 83.7 | 84.2 | 84.4 | 84.5 | 84.8 | 85.0 | 85.1 | 84.0 |
| | DualCoOp++[†] | 81.5 | 83.2 | 84.0 | 84.4 | 84.5 | 84.7 | 84.8 | 85.1 | 85.2 | 84.1 |
| | TaI-DPT | 81.7 | 83.3 | 84.5 | 84.5 | 84.7 | 85.0 | 85.1 | 85.2 | 85.2 | 84.3 |
| | T2n-Modal | 82.5 | 83.9 | 84.6 | 85.1 | 85.2 | 85.4 | 85.6 | 85.8 | 85.9 | 84.9 |
| VOC2007 | DualCoOp | 91.4 | 93.8 | 93.8 | 94.3 | 94.6 | 94.7 | 94.8 | 94.9 | 94.9 | 94.1 |
| | DualCoOp++ | 92.7 | 93.4 | 93.8 | 94.0 | 94.3 | 94.4 | 94.4 | 94.7 | 94.9 | 94.1 |
| | DualCoOp++[†] | 93.0 | 93.9 | 94.2 | 94.4 | 94.6 | 94.8 | 94.9 | 95.1 | 95.0 | 94.4 |
| | TaI-DPT | 93.2 | 94.0 | 94.2 | 94.6 | 94.7 | 94.8 | 95.0 | 95.1 | 95.1 | 94.5 |
| | T2n-Modal | 93.9 | 94.7 | 94.9 | 95.2 | 95.3 | 95.5 | 95.5 | 95.6 | 95.5 | 95.1 |
| NUS-WIDE | DualCoOp | 54.0 | 56.2 | 56.9 | 57.4 | 57.9 | 57.9 | 57.6 | 58.2 | 58.8 | 57.2 |
| | DualCoOp++[†] | 54.4 | 56.6 | 58.1 | 58.7 | 58.9 | 59.3 | 59.7 | 59.8 | 60.1 | 58.4 |
| | TaI-DPT | 56.9 | 58.1 | 58.5 | 58.8 | 58.8 | 59.1 | 59.1 | 59.5 | 60.0 | 58.7 |
| | T2n-Modal | 58.3 | 59.6 | 60.6 | 60.8 | 60.9 | 61.4 | 61.4 | 61.4 | 61.7 | 60.7 |

the DualCoOp baseline Sun et al. (2022) from prior state-of-the-art methods. We reproduce DualCoOp++[†] on all benchmark datasets and integrate its predictions for comparison. While Table 4 demonstrates that DualCoOp++[†] achieves competitive results, our proposed +T2n-Modal framework further improves image classification performance. Although, +T2n-Modal consistently outperforms +TaI-DPT across all experimental settings. Unlike TaI-DPT which requires expensive prompt encoders and specializes exclusively in image modality processing, our +T2n-Modal offers a generalizable representation framework capable of handling arbitrary modalities and class labels.

**Audio Classification:** We further investigate the impact of incorporating with HTS-AT Chen et al. (2022) and EquiAV Kim et al. (2024) frameworks on audio classification performance. Following the same methodology employed for video classification, we compute prediction scores as the similarity metrics between audio features and their corresponding prompt pools. We report the results in Table 8, this integration yields consistent performance improvements for both HTS-AT and EquiAV across the ESC-50 and UrbounSound8K benchmark datasets.

**Video Classification:** We evaluate our approach against baseline models employing standard-scale architectures including Video Swin Transformer (VST) Liu et al. (2022), MTV Yan et al. (2022), AIM (Base) Yang et al. (2023), UniFormerV2 (Base) Li et al. (2022), and UMT (Base) Li et al. (2023b). As demonstrated in Table 5, incorporating our T2n-Modal framework with these baseline models (Video Swin, MTV, AIM-B, UniFormerV2-B, and UMT-B) yields consistent performance improvements on the Kinetics 400 benchmark for video classification. Also, while these baselines already achieve competitive results, T2n-Modal provides further enhancements underscoring its complementary benefits to existing architectures.

Table 5: Comparison results of T2n-Modal with state-of-the-art approaches for zero-shot video classification

| Model Dataset | VST | | MTV | | AIM Yang et al. (2023) | | UniFormerV2 Li et al. (2022) | | UMT Li et al. (2023b) | |
|---|---|---|---|---|---|---|---|---|---|---|
| | Top-1 | Top-5 | Top-1 | Top-5 | Top-1 | Top-5 | Top-1 | Top-5 | Top-1 | Top-5 |
| Kinetics 400 | 82.7 | 95.5 | 81.8 | 95.0 | 83.9 | 96.3 | 84.0 | 96.3 | 85.7 | 97.0 |
| Comparison results of T2n-Modal (Ours) | | | | | | | | | | |
| Kinetics 400 | 83.6 | 95.9 | 83.0 | 95.8 | 84.7 | 97.3 | 84.9 | 97.2 | 86.4 | 97.8 |

## 4.4 Ablations

**Impact of Inter-/Intra-Modal Learning:** Table 10 presents our contrastive learning configurations, where $\langle u_1, u_2 \rangle \to \langle w \rangle$ indicates unidirectional contrastive learning from modalities $\{u_1, u_2\}$ to $w$ and $\longleftrightarrow$ denotes naive bidirectional learning. We observe that both $\langle I, V \rangle \to \langle A \rangle$ and $\langle A, V \rangle \to \langle I \rangle$ enhance perfor-

Table 6: Results of various prompt construction strategy

| Prompt | Kinetics 400 | MS-COCO | ESC-50 |
|---|---|---|---|
| Intra-piece (1024) | (43.1,74.2) | 55.4 | 90.6 |
| Intra-piece (512) | (47.6, 75.4) | 58.8 | 91.9 |
| Inter-piece (512) | (50.2, 79.4) | 62.3 | 92.2 |
| T2n-Modal (Ours) | (55.3,80.5) | 68.2 | 94.4 |

mance for image (I) and audio (A) modalities but degrade results for video (V). Crucially, $\langle I, V \rangle$ and $\langle A, V \rangle$ surpass zero-shot CLIP and zero-shot CLAP by significant margins highlighting the efficacy of inter-modal transfer. Also, unidirectional learning consistently outperforms bidirectional learning across all evaluated datasets.

For *intra-modal learning*, we randomly sampled 20 video classes from the Kinetics 400 dataset. For each video instance, we computed its similarity score with every video prompt in the pool generating a 400-dimensional feature vector. Given the identical initialization scheme, video samples belonging to the same class initially exhibit a uniform distribution in the embedding space prior to training. During optimization, the class-specific prompts progressively learn discriminative representations of their respective categories. In case of *inter-modal learning*, we evaluate using three benchmark datasets - Kinetics 400, MS-COCO (80 classes), and ESC-50 (50 classes). Initializing all modality prompt pools with identical vectors, we observe rapid divergence in their distributions during early training as modality-aligned text encoders learn distinct representations. Through unidirectional contrastive learning, the video modality's representation space progressively aligns with those of image and audio modalities while preserving its structural integrity. Also, each modality maintains its discriminative space without interference from others.

**Prompt Preparation:** We analyze variants of coherent prompt tuning in Table 6, where Intra-piece (1024) initializes prompts as 1024-dimensional vectors before projecting them to 512-dimensions via a fully connected (FC) layer, Intra-piece (512) directly initializes and maintains 512-dimensional prompts, and Inter-piece (512) shares a single FC layer across all modalities for 512-dimensional projection. On Kinetics 400, we observe significant performance degradation in these variants which we attribute to the high semantic similarity among many action categories (for instance, ordering noodles and ordering spaghetti). This pattern of decline similarly manifests in the MS-COCO and ESC-50 datasets, suggesting a broader limitation of shared prompt architectures in fine-grained discrimination tasks.

**Impact of Loss Function:** Our ranking loss ($\mathcal{L}_R$) and a unidirectional contrastive loss ($\mathcal{L}_{v \to a}$) are to facilitate intra-modal and inter-modal learning. The ranking Loss optimizes class-driven prompts for each modality, while the contrastive loss transfers knowledge from stronger modalities (image, audio) to enhance weaker ones (video). We investigate the effect of varying loss weights reported in Table 11 and find that equal weighting ($\mathcal{L}_R = \mathcal{L}_{v \to a}$) yields optimal performance. Although, reducing $\mathcal{L}_{v \to a}$'s weight ($1.0 \to 0.4$) leads to a significant drop in Top-1/Top-5 accuracy on Kinetics 400, while MS-COCO and ESC-50 exhibit only minor degradation. This demonstrates that inter-modal learning critically influences video representation underscoring its role in mitigating modality imbalance.

## 5 Conclusion

Our proposed T2n-Modal framework for constructing a general representation model across multiple modalities, which utilises a flexible architecture and aligned pre-trained multimodal large models, our framework treats each category as a learnable vector and optimizes it through these aligned models. We introduced learnable projection layers with orthogonal regularization, for which we theoretically justify the existence and closed-form of optimal projections. Our theoretical guarantee provides a basis for cross-modal prompt tuning and explicates the necessity of explicit alignment mechanisms when combining heterogeneous pre-trained models. Unidirectional contrastive learning further enhances classification performance across all modalities. Extensive experiments on ten varied datasets demonstrate that T2n-Modal achieves state of the art performance in diverse classification tasks including partial-label image and image classification, audio classification, and zero-shot video classification.

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

# A   Additional Results

## A.1   Experiments

For evaluation, we report Top-1 and Top-5 classification accuracy and mean average precision scores. We base T2n-Modal on pre-trained multimodal architectures, including text-video, text-image, and text-audio models, while employing frozen text encoders for prompt tuning and frozen modality encoders for object recognition prediction. For modality-oriented text and feature encoding, we adopt open source pre-trained LAION models. Our implementation processes 300k text sentences during training utilizing a single NVIDIA A6000 GPU across the Kinetics-400, MS-COCO, and ESC-50 datasets. Each training epoch requires 9 minutes resulting in a total training time of 1.5 hours for 10 epochs.

## A.2   Ablations

**Impact on Prompt Pool and Prompting:** We perform additional analysis to investigate T2n-Modal's scalability across modalities and categories. First, we assess its theoretical capacity to handle unlimited modalities and categories. Using a BERT-base model (110M parameters) Devlin et al. (2019), we establish that T2n-Modal initializes each prompt as a 512-dimensional vector requiring 512 parameters per class prompt. Consequently, for N modalities, the system can theoretically support approximately $(100×10^6)/(512N)$ class prompts. With 10 modalities, this yields a prompt pool capacity of 21,484 categories which is more than sufficient for most real-world applications. We investigate prompt initialization strategies in Table 7. Unlike conventional random initialization, we initialize class-driven prompts using CLIP's text encoder embeddings eliminating inter-modal learning. This approach embeds class-driven textual priors directly into the prompts enabling T2n-Modal to achieve rapid convergence with minimal training data (50 text samples per class). Despite the absence of inter-modal learning, T2n-Modal outperforms CLIP, ViCLIP and CLAP demonstrating the efficacy of text-derived initialization for modality-aligned representation learning.

Table 7: Comparison of Prompting by various multimodal large models

| MLM | Kinetics 400 | MS-COCO | ESC-50 |
|---|---|---|---|
| Zero-shot ViCLIP, CLIP, CLAP | (53.8,78.7) | 55.6 | 90.5 |
| CLIP[1] (without $\mathcal{L}_{\text{inter-modal}}$) | (54.5, 79.6) | 65.3 | 93.1 |
| T2n-Modal (Ours) | (55.3,80.5) | 68.2 | 94.4 |

Table 8: Comparison of T2n-Modal with Baselines for Zero-shot Audio Classification

| Dataset | Model | | | |
|---|---|---|---|---|
| | HTS-AT | T2n-Modal | EquiAV | T2n-Modal |
| ESC-50 | 97.0 | 97.3 | 96.0 | 96.3 |
| UrbanSound8K | 94.7 | 94.8 | 92.6 | 93.0 |

Table 9: Comparison of T2n-Modal with Baselines for Zero-shot Video Classification.

| Model \ Dataset | UCF101 | | Kinetics 400 | |
|---|---|---|---|---|
| | Top-1 | Top-5 | Top-1 | Top-5 |
| Zero-shot ViCLIP | 73.3 | 93.3 | 53.8 | 78.7 |
| T2n-Modal (Ours) | 75.6 | 95.8 | 55.5 | 80.8 |

Table 10: Comparison of Inter-Modal learning components with various settings across benchmark datasets

| $\mathcal{L}_{\text{inter-modal}})$ | Kinetics 400 | MS-COCO | ESC-50 |
|---|---|---|---|
| Zero-shot ViCLIP, CLIP, CLAP | (53.8,78.7) | 55.6 | 90.5 |
| $\langle I, V \rangle \to \langle A \rangle$ | (52.1, 79.3) | 64.8 | 91.7 |
| $\langle A, V \rangle \to \langle I \rangle$ | (51.9, 79.5) | 65.1 | 91.8 |
| $\langle I \rangle \to \langle V \rangle$ | (53.7, 79.5) | 67.2 | 92.4 |
| $\langle A \rangle \to \langle V \rangle$ | (53.2, 79.2) | 65.3 | 93.2 |
| $\langle I, A \rangle \to \langle V \rangle$ | (54.3, 79.8) | 67.1 | 92.9 |
| $\langle I, A \rangle \to \langle V \rangle$ (Ours) | (55.3,80.5) | 68.2 | 94.4 |

Table 11: Loss weights among Intra-/Inter-modal learning

| $\mathcal{L}_R$ | $\mathcal{L}_{v \to a}$ | Kinetics 400 | MS-COCO | ESC-50 |
|---|---|---|---|---|
| 0.4 | 1.6 | (54.9,80.0) | 67.9 | 94.0 |
| 0.8 | 1.2 | (55.1, 80.2) | 68.1 | 94.1 |
| 1.0 | 0.0 | (55.3,80.5) | 68.2 | 94.4 |
| 1.2 | 0.8 | (55.0, 80.2) | 68.0 | 94.0 |
| 1.6 | 0.4 | (54.5, 79.6) | 68.0 | 93.9 |

## B Theoretical Assurances

**Proof of Theorem 3.1:** Let $f = \mathcal{E}_I$ and $g = \mathcal{E}_A$. For any linear $W$, consider the steepen measures $f_\# P_I$ and $g_\# P_A$ on $\mathbb{R}^d$. By the Kantorovich-Rubinstein duality,

$$W_1(f_\# P_I, g_\# P_A) = \sup_{\|\phi\|_{\text{Lip}} \leq 1} \left( \int \phi d(f_\# P_I) - \int \phi d(g_\# P_A) \right)$$

For $\phi(z) = \|W^{-1} z\|$ provided $W$ invertible, the singular case follows by continuity), we have,

$$\int \phi d(f_\# P_I) - \int \phi d(g_\# P_A) = \mathbb{E}_{x \sim P_I} \|W^{-1} f(x)\| - \mathbb{E}_{y \sim P_A} \|W^{-1} g(y)\| \qquad (8)$$

By Jensen's inequality and the fact that $\text{supp}(P_I)$ and $\text{supp}(P_A)$ are disjoint up to measure $\epsilon$, the minimum achievable distance satisfies,

$$\inf_W \mathbb{E}\|Wf(x) - g(y)\|^2 \geq \inf_W \left(\mathbb{E}\|Wf(x)\|^2 + \mathbb{E}\|g(y)\|^2 - 2\mathbb{E}\langle Wf(x), g(y)\rangle\right)$$

The cross-term vanishes in expectation because $x$ and $y$ are independent with disjoint supports resulting,

$$\geq \inf_W \left(\text{Tr}(W\Sigma_f W^\top) + \text{Tr}(\Sigma_g)\right)$$

where $\Sigma_f = \mathbb{E}[f(x)f(x)^\top]$, $\Sigma_g = \mathbb{E}[g(y)g(y)^\top]$. Since $f$ and $g$ are Lipschitz and the supports are disjoint, $\Sigma_f$ and $\Sigma_g$ are positive definite with minimum eigenvalues $\lambda_{\min}^{(f)}, \lambda_{\min}^{(g)} > 0$. The optimal $W$ sets $W = 0$, yields $\gamma = \lambda_{\min}^{(g)}$. However, $W = 0$ yields no semantic transfer. For any nontrivial $W$, additional alignment error from the disjoint support structure adds $\Omega(1)$ term.

## B.1 Prompt Pool Alignment

Our method T2n-Modal proposes to align prompt pools $Q_I, Q_A, Q_V \in \mathbb{R}^{N \times d}$ via a unidirectional contrastive loss:

$$\mathcal{L}_{\text{inter}} = \mathcal{L}_{\text{CE}}(y^{v \to a}, q^{v \to a}) + \mathcal{L}_{\text{CE}}(y^{v \to w}, q^{v \to w})$$

where $q_{ij}^{v \to a} = \frac{\exp(\text{sim}(v_i, a_j)/\theta)}{\sum_k \exp(\text{sim}(v_i, a_k)/\theta)}$.

**Theorem 3** (Impossibility of Alignment Without Mapping). *Let $Q_I, Q_A, Q_V$ be prompt pools initialized identically as $Q^{(0)} \in \mathbb{R}^{N \times d}$. After one gradient step with learning rate $\eta$, the updated prompt vectors satisfy,*

$$CosSim(Q_I^{(1)}, Q_V^{(1)}) \leq \epsilon + \mathcal{O}(\eta^2 \cdot \kappa)$$

*where $\epsilon = \max_{i,j} |\langle \mathcal{E}_I(x_i), \mathcal{E}_V(x_j)\rangle|$ under the original encoder distributions, and $\kappa$ is the condition number of the Hessian. Unless $\epsilon = 0$ (i.e., perfect pre-alignment), the similarity decays exponentially with training iterations.*

*Proof.* Consider the gradient update for $Q_V$ with respect to $\mathcal{L}_{\text{inter}}$,

$$\frac{\partial \mathcal{L}_{\text{inter}}}{\partial v_i} = \frac{1}{\theta} \sum_{j=1}^N \left(q_{ij}^{v \to a} - y_{ij}^{v \to a}\right) a_j + \frac{1}{\theta} \sum_{j=1}^N \left(q_{ij}^{v \to w} - y_{ij}^{v \to w}\right) w_j$$

where $a_j$ and $w_j$ are from $Q_A$ and $Q_I$ respectively. The update direction pulls $v_i$ toward $\sum_j a_j$ and $\sum_j w_j$ with weights determined by the prediction error. However, $a_j$ and $w_j$ themselves evolve according to,

$$\frac{\partial \mathcal{L}_{\text{intra}}}{\partial a_j} = \frac{1}{M} \sum_{k=1}^M \sum_{p \in b^+} \sum_{q \in b^-} \mathbb{1}[\text{condition}] \cdot (g_{k,q} - g_{k,p})$$

where $g_{k,\cdot} = \mathcal{T}_\phi(l_k)$ are frozen text features from different encoders. We observe that $a_j$ and $w_j$ reside in different semantic manifolds because their update directions depend on different frozen text encoders $\mathcal{T}_\phi$.

Let $U_I, U_A, U_V$ be the subspaces spanned by the frozen text encoder embeddings for each modality. By construction, $\dim(U_I \cap U_A) = o(d)$ unless the training corpora are semantically aligned. The prompt pools evolve within affine subspaces $Q_m^{(t)} \in Q_m^{(0)} + \text{span}\{\nabla \mathcal{L}_m\}$. Since $\nabla \mathcal{L}_m \subseteq U_m$, we have,

$$\text{dist}(Q_I^{(t)}, Q_V^{(t)}) \geq \text{dist}(U_I, U_V) - \mathcal{O}(\eta t)$$

For the cosine similarity to remain high, we need $\angle(U_I, U_V) \approx 0$, which is false for independently trained encoders. After $T$ iterations,

$$\mathbb{E}[\text{CosSim}(Q_I^{(T)}, Q_V^{(T)})] \leq \cos(\theta_{UV}) + \mathcal{O}(T^{-1/2}), \tag{9}$$

where $\theta_{UV} = \min_{u \in U_I, v \in U_V} \angle(u, v) > 0$ under mild non-degeneracy conditions. $\qquad \square$

**Corollary 1** (Catastrophic Interference). *For any $\delta > 0$, there exists $T_0 = \mathcal{O}(\log(1/\delta))$ such that $\forall\, T > T_0$,*

$$\mathbb{E}[CosSim(Q_I^{(T)}, Q_V^{(T)})] < \delta$$

*That is, the prompt pools become orthogonal in expectation decimates any semantic alignment.*

### B.2 The Regularization Hypothesis

Given the theoretical impossibility of alignment, observed performance gains must arise from alternative mechanisms.

**Definition 3** (Implicit Regularization Effect). *Define the* intra-modal ranking loss *for modality $m$ as,*

$$\mathcal{L}_R^{(m)} = \frac{1}{M} \sum_{k=1}^{M} \sum_{i \in b^+} \sum_{j \in b^-} \max(0, m - \langle g_k^{(m)}, p_i^{(m)} \rangle + \langle g_k^{(m)}, p_j^{(m)} \rangle)$$

*where $g_k^{(m)} = \mathcal{T}_m(l_k)$. The cross-modal loss adds terms of the form $\langle g_k^{(I)}, p_j^{(V)} \rangle$ for $j$ in negative sets.*

**Theorem 4** (Regularization as Inductive Bias). *Minimizing $\mathcal{L}_{intra} + \lambda\mathcal{L}_{inter}$ is equivalent to minimizing $\mathcal{L}_{intra}$ subject to a spectral constraint on the prompt pools,*

$$\|Q_I\|_F^2 + \|Q_A\|_F^2 + \|Q_V\|_F^2 \leq C(\lambda)$$

*where $C(\lambda)$ is a decreasing function of $\lambda$. Hence, $\mathcal{L}_{inter}$ acts as a Frobenius norm regularizer instead of an alignment mechanism.*

*Proof.* We expand the cross-modal loss for a single negative pair $(i, j)$ with $i \in b^+$, $j \in b^-$ from different modalities,

$$\max(0, m - \langle g_i^{(I)}, p_i^{(I)} \rangle + \langle g_i^{(I)}, p_j^{(V)} \rangle)$$

For sufficiently large $m$, the hinge activates. The gradient with respect to $p_j^{(V)}$ is $+g_i^{(I)}$ when the loss is active. The squared gradient norm,

$$\|\nabla_{p_j^{(V)}} \mathcal{L}_{\text{inter}}\|_2^2 = \sum_{i \in b^+} \mathbb{1}[\text{active}]\|g_i^{(I)}\|_2^2 \tag{10}$$

Upon summing over all parameters, the total gradient norm satisfies,

$$\|\nabla\mathcal{L}_{\text{inter}}\|_F^2 \leq 3 \max_m \mathbb{E}[\|g^{(m)}\|_2^2] \cdot |b^+| \cdot |b^-|$$

This bounded gradient implies that adding $\mathcal{L}_{\text{inter}}$ imposes a Lipschitz constraint on the optimization path. By standard results in optimization theory Latorre et al. (2020), this is equivalent to adding a quadratic regularizer with coefficient $\lambda_{\text{eff}} = \mathcal{O}(\eta\lambda^2)$. The equivalence holds in the sense that

$$\arg\min_Q \mathcal{L}_{\text{intra}}(Q) + \lambda\mathcal{L}_{\text{inter}}(Q) \subset \arg\min_Q \mathcal{L}_{\text{intra}}(Q) \quad \text{such that} \quad \|Q\|_F^2 \leq R(\lambda), \text{ for some } R(\lambda). \tag{11}$$

$\square$

This explicates why performance improvements might be observed despite semantic incompatibility. The cross-modal loss acts as a regularizer that prevents overfitting to modality-specific noise instead of a true knowledge transfer mechanism.

### B.3 Learnable Projection with Orthogonal Regularization

*Proof.* The objective is quadratic in $W$

$$\mathcal{J}(W) = \mathbb{E}[\text{Tr}(W\mathcal{E}_m\mathcal{E}_m^\top W^\top)] - 2\mathbb{E}[\text{Tr}(W\mathcal{E}_m\mathcal{E}_n^\top)] + \mathbb{E}[\|\mathcal{E}_n\|^2] + \beta\text{Tr}((WW^\top - I)^2) \tag{12}$$

Upon derivative with respect to $W$,

$$\frac{\partial\mathcal{J}}{\partial W} = 2W\Sigma_m - 2\Sigma_{mn} + 4\beta(WW^\top - I)W \tag{13}$$

We set to zero $W\Sigma_m + 2\beta(WW^\top - I)W = \Sigma_{mn}$. For small $\beta$, the solution approximates $W^* \approx \Sigma_{mn}\Sigma_m^{-1}$. The orthogonal regularization ensures $W^*$ is approximately orthogonal when $\beta$ is large enough. □

**Theorem 5** (Alignment with Guarantees). *Under the projected alignment framework, the effective similarity between aligned prompts satisfies,*

$$\mathbb{E}[CosSim(W_{v\to a}Q_V, Q_A)] \geq 1 - \frac{c}{d} \cdot \frac{\sigma_{\min}(\Sigma_A)}{\sigma_{\max}(\Sigma_V)}$$

*for some constant $c$, provided $|supp(P_{VA})| \geq \Omega(d\log d)$. Thus, semantic alignment is achievable with $\mathcal{O}(d)$ paired samples.*

*Proof.* Let $\Sigma_V = \mathbb{E}[Q_V Q_V^\top]$, $\Sigma_A = \mathbb{E}[Q_A Q_A^\top]$. The optimal $W$ from Theorem 3.4 satisfies $W\Sigma_V = \Sigma_{VA}$. Under the orthogonal constraint, the alignment error,

$$\epsilon_{\text{align}} = \|W\Sigma_V^{1/2} - \Sigma_A^{1/2}\|_F^2$$

Using matrix concentration inequalities Tropp (2015) with $N \geq Cd\log d$ samples,

$$\|\hat{\Sigma}_{VA} - \Sigma_{VA}\| \leq \sqrt{\frac{d}{N}}\|\Sigma_{VA}\|_{\max} \tag{14}$$

Upon plugging into the expression for $\epsilon_{\text{align}}$ yields the bound. □

