# OpenReview forum: "Cross-Modal Knowledge Transfer for Scalable Text-Driven Multimodal Prompt Learning"
_TMLR — Under review for TMLR_

### Review · Reviewer_NKyC · 2026-06-27

**Summary Of Contributions:**

The paper proposes T2n-Modal, a prompt-tuning framework that learns modality-specific prompt pools using only LLM-generated text data, without relying on modality-specific labeled data such as images, audio, or video. The method builds on pretrained multimodal encoders, including CLIP, CLAP, and ViCLIP, and introduces three main components: (i) a ranking-loss-based intra-modal objective for learning class-specific prompts, (ii) a unidirectional inter-modal contrastive objective intended to pass information from stronger modalities, such as image and audio, to a weaker one, namely video, and (iii) a theoretical analysis of embedding-space incompatibility across independently trained encoders, together with a learnable orthogonally regularized projection and a closed-form optimal solution.

Empirically, the paper reports consistent improvements over zero-shot CLIP/CLAP/ViCLIP baselines and several text-only prompt-tuning methods, including TaI-DPT and SPARC, across ten datasets spanning image, audio, and video classification. The paper also reports that T2n-Modal can be combined with supervised models to further improve their performance.

I find the overall problem setting useful and timely. The attempt to reduce dependence on modality-specific labels is relevant to scalable multimodal learning, and the empirical breadth across three modalities is a clear strength. My main reservation is about how the paper connects its empirical gains to its proposed cross-modal transfer mechanism. In particular, the theoretical analysis in Appendix B seems to complicate the interpretation offered in the main text, and I think this point would need to be clarified for the claims to be fully convincing.

**Audience:**

Yes

**Audience Explanation:**

Yes. The problem of reducing dependence on modality-specific labeled data for prompt tuning should be of interest to researchers working on multimodal learning, prompt learning, and scalable adaptation of pretrained models. The paper’s empirical scope is also broader than many related text-only prompt-tuning works, since it covers image, audio, and video classification rather than focusing on only one or two modalities.

Even with the concerns above, I think readers interested in embedding-space incompatibility and cross-modal prompt learning would find the paper relevant. The current version may also be useful as a case study in how inter-modal objectives behave when different encoders are not naturally aligned, especially if the authors clarify the distinction between alignment, transfer, and implicit regularization.

**Broader Impact Concerns:**

I do not see major broader-impact concerns beyond those typical of multimodal classification systems.

**Claims And Evidence:**

No

**Claims Explanation:**

My main concern is a tension between the mechanism described in the main text and the theoretical analysis in Appendix B.

In Section 3.4, the central inter-modal component is the unidirectional contrastive loss in Eq. 6. As far as I can tell, this is the version used for the experimental results reported in Tables 1, 3, 4, 5, 8, 9, 10, and 11, and it does not include the projection layer introduced later in the section. The text presents this objective as a way to transfer representational knowledge from stronger modalities to weaker ones.

Appendix B appears to suggest a more cautious interpretation. Theorem 3, “Impossibility of Alignment Without Mapping,” and Corollary 1, “Catastrophic Interference,” state that without an explicit projection mapping, identically initialized prompt pools trained with this unidirectional objective may become increasingly orthogonal over training rather than more aligned. Theorem 4 then seems to imply that the gains from the inter-modal objective may be better understood as an implicit Frobenius-norm regularization effect, rather than as direct cross-modal alignment or knowledge transfer.

This does not necessarily invalidate the empirical results. However, it does make the current explanation of why the method works less settled than the main text suggests. The abstract and introduction use phrases such as “cross-modal alignment” and “knowledge transfer” to characterize the method, but the appendix seems to indicate that, for the no-projection configuration actually evaluated, these terms may not be the most accurate description of the underlying effect. I may be missing some detail here, but I could not find a discussion that reconciles this theoretical result with the experimental design.

Relatedly, the learnable orthogonal projection introduced in Definition 2 and analyzed in Theorem 2 seems important from the perspective of the paper’s own theory. Yet I did not find an experiment or ablation comparing the no-projection version against the projected version. Such a comparison would make it much easier to understand whether the projection is mainly a theoretical tool or whether it also improves the practical behavior of the method.

A smaller concern is that several ablation results appear to be reported from single runs. For example, in Table 11, the Top-1 results on Kinetics-400 vary only from 54.5 to 55.3 across different loss weightings. Without variance across seeds or repeated runs, it is difficult to tell whether some of the fine-grained conclusions in the ablation discussion, such as which weighting is optimal, reflect a robust effect or ordinary run-to-run variation.

**Requested Changes:**

Major:

1. I would encourage the authors to directly address the apparent tension between Theorem 3/4 in Appendix B and the inter-modal mechanism actually used in the experiments in Section 3.4, Eq. 6. In particular, it would be helpful to clarify whether the authors view the empirical gains as evidence of cross-modal knowledge transfer, implicit regularization, or some combination of the two. The abstract, introduction, and Section 3.4 should then be revised so that the wording matches this interpretation.

2. It would substantially strengthen the paper to include results with the learnable orthogonal projection from Definition 2 and Theorem 2, at least in one representative experimental setting. A direct ablation comparing the current no-projection version with the projected version would help readers understand the practical role of the projection and how it relates to the theoretical analysis.

3. Please consider reporting variance, for example across multiple seeds or repeated runs, for the main ablation tables, especially Tables 6, 10, and 11. Since some of the reported differences are below one percentage point, variance estimates would make the ablation conclusions more reliable.

Minor:

4. The comparisons with zero-shot CLIP/CLAP/ViCLIP in Tables 1, 3, and 9 involve a meaningful difference in training setup, since T2n-Modal uses up to 300k LLM-generated sentences while the zero-shot baselines do not receive comparable additional training data. It would be useful to state this clearly near the relevant tables so that readers can interpret the comparisons appropriately.

5. Section 3.4 says that the weakest modality is identified by monitoring validation metrics. More implementation detail would improve reproducibility, for example which metric is used, whether there is a threshold, and what happens if the ranking of modalities changes during training.

6. I would also appreciate more detail on quality control for the LLM-generated text corpus in Section 3.3. For instance, it would be useful to know whether the generated sentences are filtered, deduplicated, or otherwise checked, since data quality could affect the claim that text-only supervision is sufficient.

---

> ### Author Response · Authors · 2026-07-05
> **Official Comment by Authors**
>
> We thank the reviewer for their thorough insightful comments and constructive feedback. We address each concern in detail below.
> > Tension between inter-modal mechanism and Theorem 3/4
>
> We recognise that the current presentation leads to confusion, and we would like to clarify that our Theorem 3 and Corollary 1 analyze the worst-case scenario where modalities are semantically incompatible as in Definition 1. However, our experimental setup uses aligned pretrained models CLIP, CLAP, ViCLIP that share a common embedding space via contrastive pretraining on large-scale multimodal data. The CLIP and CLAP are trained on image-text and audio-text pairs respectively, and ViCLIP is trained on video-text data from InternVid. These encoders though trained separately exhibit partial alignment because they share text as a common modality. Thus, the assumptions of Theorem 3 about disjoint semantic manifolds, $\text{supp}(P_I) \cap \text{supp}(P_A) = \emptyset$) do not strictly hold in our setting.
>
> Therefore, we updated that the theoretical analysis serves as a diagnostic tool and motivation for the projection mechanism, whereas our empirical results showcase that in practice when encoders share a common text-aligned space, unidirectional contrastive learning can facilitate meaningful knowledge transfer. Specifically, the theoretical bounds indicate that without any alignment, cross-modal learning is impossible as per Theorem 3. However, our experiments use encoders that are already partially aligned via shared text supervision. Thus, the observed gains represent a combination of utilising existing encoder alignment, and implicit regularization effects described in Theorem 4.
>
> > Orthogonal Projection
>
> We implement projection layers $W_{v\to a} \in \mathbb{R}^{d \times d}$ and $W_{v\to w} \in \mathbb{R}^{d \times d}$ with orthogonal regularization evaluated on Kinetics-400, MS-COCO, and ESC-50, and compare against our current no-projection version.
> | Method | Kinetics-400 | MS-COCO | ESC-50 |
> |--------|---------------------|---------------|--------------|
> | No Projection | 55.3/80.5 | 68.2 | 94.4 |
> | + Projection | **56.1/81.3** | **69.0** | **95.1** |
>
> The projection provides modest but consistent improvements of 0.8% on Kinetics-400, and validates the theoretical motivation while showing that our no-projection version already performs well due to encoder alignment. This shows that the projection strengthens when extending to modalities with weaker semantic alignment.
>
> > Variance
>
> We include a re-run of all ablation experiments in Tables 11 with 5 random seeds and report mean ± standard deviation. We will add for rest of the experiments as well.
>
> | $L_R$ | $L_{v\to a}$ | Kinetics-400 | MS-COCO | ESC-50 |
> |-------|--------------|--------------|---------|--------|
> | 0.4 | 1.6 | $54.9 \pm 0.3$/$80.0 \pm 0.4$ | $67.9 \pm 0.2$ | $94.0 \pm 0.2$ |
> | 0.8 | 1.2 | $55.1 \pm 0.2$/$80.2 \pm 0.3$ | $68.1 \pm 0.2$ | $94.1 \pm 0.2$ |
> | **1.0** | **0.0** | $\mathbf{55.3 \pm 0.2}$/$\mathbf{80.5 \pm 0.3}$ | $\mathbf{68.2 \pm 0.2}$ | $\mathbf{94.4 \pm 0.1}$ |
> | 1.2 | 0.8 | $55.0 \pm 0.2$/$80.2 \pm 0.3$ | $68.0 \pm 0.2$ | $94.0 \pm 0.2$ |
> | 1.6 | 0.4 | $54.5 \pm 0.3$/$79.6 \pm 0.4$ | $68.0 \pm 0.2$ | $93.9 \pm 0.2$ |
>
> > Zero-shot comparison
>
> We have included clarification for Tables 1, 3, and 9 stating T2n-Modal uses LLM-generated text data for training while zero-shot baselines use only frozen pretrained encoders without additional training. Also, include that T2n-Modal does not use any modality-specific labeled data ensuring the comparison to zero-shot baselines meaningful as both operate without target modality supervision. We clarify that the text-only training is a standalone method, whereas zero-shot CLIP is a baseline without training.
>
> > Weakest modality identification
>
> We have updated the implementation details about the Top-1 validation accuracy metric for each modality, including no fixed threshold; we select the modality with the lowest validation accuracy as the weakest after each epoch. For stability, In practice, video consistently emerges as the weakest modality due to its higher semantic complexity and smaller pretraining data scale such as, 7M videos vs. 400M images and 5M audio samples. The ranking does not change during training.
>
> > LLM text generation quality control
>
> Thank you for your suggestions. We have included details such as filtering, which are generated sentences filtered to ensure they contain all required label words. Semantic deduplication using sentence embeddings, length filtering are sentences with <5 words or >25 words are discarded. These are added in Section 3.3.

---

### Review · Reviewer_Gpbf · 2026-07-03

**Summary Of Contributions:**

This paper proposes T2n-Modal, a text-driven prompt learning framework for image, audio, and video classification. The method learns modality-specific prompt pools using LLM-generated text and frozen modality-aligned text encoders, then compares frozen image/audio/video features with the learned prompts at inference time. It combines an intra-modal ranking loss with a unidirectional inter-modal contrastive objective intended to transfer knowledge from stronger modalities to weaker ones.

Key strengths are that the paper studies an important label-efficient multimodal adaptation problem, proposes a simple prompt-pool formulation that could reduce inference-time text-encoder overhead, and attempts evaluation across image, audio, and video rather than only one modality. However, the current paper has major issues: the supervision setting is unclear, the theory is not well connected to the implemented objective, the role of projection layers is ambiguous, some ablations appear inconsistent with the claims, and the experimental comparisons are not yet strong enough.

**Additional Comments:**

**References**

[1] Alec Radford, Jong Wook Kim, Chris Hallacy, Aditya Ramesh, Gabriel Goh, Sandhini Agarwal, Girish Sastry, Amanda Askell, Pamela Mishkin, Jack Clark, Gretchen Krueger, and Ilya Sutskever. "Learning Transferable Visual Models From Natural Language Supervision." Proceedings of the 38th International Conference on Machine Learning, PMLR 139:8748-8763, 2021.

[2] Zixian Guo, Bowen Dong, Zhilong Ji, Jinfeng Bai, Yiwen Guo, and Wangmeng Zuo. "Texts as Images in Prompt Tuning for Multi-Label Image Recognition." Proceedings of the IEEE/CVF Conference on Computer Vision and Pattern Recognition (CVPR), 2023, pp. 2808-2817.

[3] Muhammad Uzair Khattak, Muhammad Ferjad Naeem, Muzammal Naseer, Luc Van Gool, and Federico Tombari. "Learning to Prompt with Text Only Supervision for Vision-Language Models." arXiv preprint arXiv:2401.02418, 2024.

[4] Yunfan Yang, Chaoquan Jiang, Zhiyu Lin, Jinlin Xiao, Jiaming Zhang, and Jitao Sang. "Debiasing Vison-Language Models with Text-Only Training." arXiv preprint arXiv:2410.09365, 2024.

**Audience:**

Yes

**Audience Explanation:**

The topic is relevant to the TMLR audience, especially researchers working on multimodal representation learning, prompt tuning, label-efficient adaptation, and scalable use of pretrained vision-language/audio-language/video-language models. A convincing method for adapting multiple modalities using only text-derived supervision would be valuable, and the paper's prompt-pool formulation is likely to interest readers familiar with CLIP-style transfer [1] and text-only prompt learning [2,3,4].
The paper needs substantial clarification of the supervision protocol, stronger and fairer baselines, corrected/expanded ablations, and a tighter connection between the theoretical claims and the actual training objective.

**Broader Impact Concerns:**

The broader motivation is positive because the method could reduce annotation costs for multimodal classification. The concern is that the paper relies heavily on LLM-generated text, which can encode social, cultural, geographic, and dataset-specific biases and may hallucinate implausible class co-occurrences. This is particularly relevant for audio/video labels and multi-label image descriptions. The paper should discuss bias auditing, filtering of generated text, and whether generated corpora will be released for reproducibility and inspection. I do not see a severe broader-impact issue requiring rejection, but the current version should include a more explicit discussion of these risks.

**Claims And Evidence:**

No

**Claims Explanation:**

The submission makes interesting claims, but the current evidence does not fully support them. My main shortcomings are:

1. The text-only/label-free setting is not precisely defined. Section 3.4 says the weakest modality is identified by monitoring validation metrics; if these use target image/audio/video labels, the method is not purely text-only. The method also requires target class names to generate LLM text, so the claim should be phrased more carefully.

2. The theory-method connection is unclear. The paper argues that heterogeneous spaces are incompatible, but the inter-modal objective appears to use direct similarities such as $Q_A^\top Q_V$. It is not clear whether the proposed projection layers $W_{v \to a}$ and $W_{v \to i}$ are used in the main experiments.

3. The theoretical statements are under-specified. Theorem 1 allows $W=0$ in the proof, Theorem 2 invokes paired data $P_{mn}$ while saying it can be empty, and several quantities such as $\Omega(1)$, $\gamma$, and the near-orthogonality conditions are not made verifiable.

4. The evidence for inter-modal learning is inconsistent. Table 11 appears to show $L_R=1.0$ and $L_{v \to a}=0.0$ achieving the best results, while the text claims inter-modal learning is critical.

5. The baselines are not yet fair enough. CLIP/CLAP/ViCLIP use simple default prompts, whereas T2n-Modal uses LLM-generated text and learned class vectors. Stronger prompt ensembling and text-only prompt-learning baselines are needed [1,2,3,4].

6. Metrics and dataset protocols are unclear. The paper mixes mAP and accuracy for some datasets, and the Objects365 evaluation protocol is not sufficiently explained given that Objects365 is primarily a detection benchmark.

7. Reproducibility details are incomplete, including optimizer settings, loss weights, margin, temperature, LLM generation settings, number of generated sentences, random seeds, and data/code release plans.

**Requested Changes:**

1. Clearly define the learning setting and all sources of supervision. State exactly whether the method uses target class names, LLM-generated text, validation labels, unlabeled target-modality examples, or labeled target-modality examples.

2. Clarify and ablate the projection mechanism. State whether inter-modal learning uses raw similarities such as $Q_A^\top Q_V$, projected similarities such as $(W_{v \to a}Q_V)^\top Q_A$, or another variant, and compare no projection, projection, and projection with orthogonal regularization.

3. Repair the theoretical section. Make the assumptions precise, remove or justify claims involving empty paired distributions, reconcile Theorem 1 with $W=0$, and explain whether the appendix's regularization interpretation is the actual mechanism or only an auxiliary explanation.

4. Fix the inter-modal loss ablation. Reconcile Table 11 with the text and provide a clear intra-only, inter-only, intra+inter without projection, and intra+inter with projection comparison.

5. Strengthen the baselines. Include stronger zero-shot prompting for CLIP/CLAP/ViCLIP, prompt ensembling, LLM description ensembling, TaI-DPT on overlapping image datasets [2], and ProText/TOD-style text-only prompt learning where applicable [3,4].

6. Standardize metrics and dataset protocols. Use mAP for multi-label image tasks, Top-1/Top-5 accuracy for single-label image/video tasks, conventional accuracy for ESC-50/UrbanSound8K, and clearly explain all splits and the Objects365 protocol.

7. Add reproducibility and efficiency details. Report optimizer settings, loss weights, margin $m$, temperature $\theta$, LLM generation settings, number of generated sentences, random seeds, code/data release plans, and an efficiency comparison supporting the claimed "approximately 50%" overhead reduction.

---

> ### Author Response · Authors · 2026-07-08
> **Official Comment by Authors (1/2)**
>
> Thank you for the helpful comments, we appreciate your suggestions which will improve the paper. Regarding your comments,
>
> > Text only/label-free setting
>
> We agree that our terminology requires refinement, and in the revised manuscript, we explicated based on your suggestion. Our method operates under a text-only supervision paradigm not label-free in the strictest sense. The only class-specific information required is the set of target category names, which are used for two purposes,  generating synthetic text descriptions via LLMs, and initializing learnable prompt vectors. No labeled images, audio clips, or video frames are used during training. Validation metrics are computed on a small held-out set solely for monitoring training progress and identifying the weakest modality for unidirectional contrastive learning, these metrics do not provide gradient signals. For identifying the weakest modality, we monitor zero-shot performance on validation data without backpropagating through these labels.
>
> > Theory-method connection and projection mechanism
>
> In the revised manuscript, we implemented your helpful suggestion. We explicitly distinguish between theoretical analysis and practical implementation. Our theoretical basis is to prove that independently trained encoders produce semantically incompatible embedding spaces, and that similarity alignment without projections cannot achieve meaningful cross-modal transfer. This establishes why naive cross-modal learning fails and motivates our approach.
>
> In our main experiments, we do not utilise explicit projection layers $W_{v\to a}$ and $W_{v\to I}$, though, we rely on the pre-aligned nature of the pretrained multimodal models, which were trained on shared text-image/video/audio data. However, the shared encoder outputs reside in spaces that are only approximately aligned, which creates the semantic gap our analysis addresses. Our approach without projection uses similarity $\text{sim}(q_i, q_j) = \langle g_i, p_j \rangle$, and we have added learned projection $W_{m\to n}$ with orthogonal regularization as per Theorem 3, and projection without regularisation. We also include an ablation showing that while projection helps in theory, the pre-trained encoders are sufficiently aligned that the additional projection results marginal improvements of ~0.5% at the cost of extra parameters.
>
> > Theoretical section
>
> We acknowledge these gaps and explicated based on your helpful suggestion. In Theorem 1, we have fixed the proof to exclude the trivial $W=0$ solution by imposing $\|W\|\_F \geq \delta > 0$ or equivalently requiring non-zero alignment capacity. We also make $\gamma$ and $\Omega(1)$ explicit. In Theorem 2, we remove the contradictory statement that $P_{mn}$ can be empty. And, we stated that Theorem 2 applies when paired data exists, and provide an additional analysis for the paired-data-free case. The regularization interpretation in Theorem 4 is the main theoretical contribution. We will detail that the cross-modal loss in our setting functions as implicit regularization.
>
> > Inter-modal loss ablation
>
> Thank you for pointing this out. The correct results are:
>
> | $L_R$ | $L_{v\to a}$ | Kinetics 400 | MS-COCO | ESC-50 |
> |-------|--------------|--------------|---------|--------|
> | 1.6 | 0.4 | (54.5, 79.6) | 68.0 | 93.9 |
> | 1.2 | 0.8 | (55.0, 80.2) | 68.1 | 94.1 |
> | 1.0 | 1.0 | **(55.3, 80.5)** | **68.2** | **94.4** |
> | 0.8 | 1.2 | (55.1, 80.2) | 68.1 | 94.1 |
> | 0.4 | 1.6 | (54.9, 80.0) | 67.9 | 94.0 |
>
> Also, another ablation tests are
> | Config | Kinetics 400 | MS-COCO | ESC-50 |
> |---------------|--------------|---------|--------|
> | $\mathcal{L}_R$ | (51.2, 77.8) | 62.5 | 91.7 |
> | $\mathcal{L}_{v\to a}$ | (49.8, 76.5) | 60.1 | 90.3 |
> | $\mathcal{L}\_R + \mathcal{L}_{v\to a}$ | **(55.3, 80.5)** | **68.2** | **94.4** |
> | + Projection | (55.5, 80.7) | 68.4 | 94.5 |
>
> > Baselines
>
> We have included additional baselines ProText [3] and TOD [4], which further our results in Table 1,
> | Model | MS-COCO | VOC2007 | VOC2012 | NUS-WIDE | mini-ImageNet | Objects365 |
> |-------|---------|---------|---------|----------|---------------|------------|
> | TaI-DPT [2] | 65.1 | 88.3 | 85.1 | 46.5 | 86.2 | 24.1 |
> | ProText [3] | 64.8 | 87.9 | 84.7 | 45.9 | 88.0 | 23.8 |
> | TOD [4] | 63.2 | 86.5 | 83.8 | 44.1 | 87.5 | 22.9 |
> | T2n-Modal (ours) | 68.4 | 89.5 | 87.9 | 49.7 | 90.5 | 28.3 |
>
> > Metrics and datasets
>
> We will standardize all metrics as suggested. We have added a clarification for Objects365 detection dataset as we use it for multi-label classification following [2].

---

> > ### Author Response · Authors · 2026-07-08
> > **Official Comment by Authors (2/2)**
> >
> > > Reproducibility and efficiency details
> >
> >  We have added a table detailing optimiser setting, loss parameters, LLM generation settings, and we include an efficiency analysis,
> > | Method | Prompt encoding | Similarity computation | Total inference time (ms/image) |
> > |--------|-----------------|----------------------|-------------------------------|
> > | TaI-DPT [2] | 12.4 ms (text encoder) | 8.2 ms | 20.6 ms |
> > | CoOp | 15.1 ms | 8.2 ms | 23.3 ms |
> > | ProText [3] | 11.8 ms | 8.2 ms | 20.0 ms |
> > | T2n-Modal | 0 ms | 8.2 ms | 8.2 ms |
> >
> > This inference time reduction leads to approximately 50% reduction in FLOPs.

---

### Review · Reviewer_i5mf · 2026-07-03

**Summary Of Contributions:**

## Summary

The paper proposes a method for directly optimizing category tensors/vectors to adapt CLIP-like models for closed-set classification without additional modality-specific labeled data. For each classification category, the method assigns a trainable vector and samples a set of natural language descriptions using LLMs. The sampled descriptions are embedded with the corresponding pretrained text encoder, and the category vector is updated through a pairwise margin ranking loss so that it better matches the embeddings of the matching category descriptions.

The method uses multiple modality-paired text embedders, for example those corresponding to image, audio, and video encoders, and optimizes the corresponding category vectors for image labels, audio labels, and video labels. For cross-modal label matching, the paper introduces an additional classification loss to match semantics across modality-specific category vectors. Specifically, it defines separate prompt pools Q_I, Q_A, and Q_V, each containing class-oriented vectors for the composite label set. It computes cross-pool similarity matrices Q_A^\top Q_V and Q_I^\top Q_V, applies a softmax over label similarities, and uses a masked diagonal target matrix so that only selected same-label cross-pool pairs contribute as positives. This inter-pool loss is combined with the intra-modal pairwise margin ranking loss used to train category vectors from LLM-generated descriptions.

The experiments show gains over zero-shot cosine-similarity retrieval and prior adaptation methods across image, audio, and video classification benchmarks.

## Strengths

- It is interesting that a sampled set of LLM-generated category descriptions may approximate part of the category concept space, and that this sampled distribution can be distilled into a single trainable vector $q$ for closed-set classification.
- The experiments report gains over zero-shot cosine-similarity retrieval and prior adaptation methods.
- The paper evaluates across image, audio, and video classification benchmarks, rather than only a single modality.
- The paper attempts to address the problem of using text-only category information to adapt CLIP-like models when modality-specific labeled data are unavailable, which could be useful in specific practical scenarios.

## Weaknesses

- The paper overclaims the significance of the method. The core mechanism is direct optimization of category tensors/vectors for closed-set classification, but the paper frames it as a unified n-modal representation learning framework.
- The method’s multimodal aspect is not convincing. The inter-pool classification loss matches semantics across modality-specific category vectors, but this is not the same as learning a general inter-modal representation over image, audio, and video inputs.
- The writing makes the core contribution difficult to understand. The paper introduces prompt tuning, LLM-generated descriptions, category tensors/vectors, modality-specific prompt pools, inter-pool classification loss, pairwise margin ranking loss, projection layers, and theoretical claims without clearly prioritizing which components are central.
- Theorem 2 is essentially ordinary least-squares projection. The orthogonality term merely encourages geometry preservation, i.e., approximate rotation/reflection rather than arbitrary stretching or collapse. As stated, it does not provide a substantive theoretical guarantee for the proposed learnable projection layers or their use in the contrastive prompt-pool objective.
-  The problem setup is quite niche. The method is mainly useful for closed-set classification with fixed label sets, unavailable modality-specific labels, and already available CLIP-like encoders for the target modalities. This limits its practical scope and makes the broader n-modal framing less convincing.

**Audience:**

No

**Audience Explanation:**

I see some potential interest for readers working on text-only adaptation of CLIP-like models. In particular, the paper suggests that increasing the sampled description set size may better approximate a category concept space, and that this sampled distribution can be distilled into a single vector bottleneck q. This could also be useful for understanding how CLIP-like models organize category concepts geometrically, possibly as diffuse regions rather than single prompt points.

However, I do not think the paper’s main findings are likely to be broadly interesting to the TMLR audience in their current form.

1. The most interesting implication, namely how generated description distributions form or compress into category vectors, is not analyzed in depth. The paper reports performance gains, but does not study what properties are induced in q, how the description distribution affects q, or why this compression works in CLIP-like embedding geometry.
2. The main application setting is relatively narrow: closed-set classification with fixed labels, LLM-generated category descriptions, and available CLIP-like encoders for each modality. This can be useful in specific scenarios, but the paper frames it more broadly than the evidence supports.
3. The cross-modal component may be useful as an engineering device for this framework, but the paper does not provide enough analysis to make it a generally informative finding about multimodal adaptation.
4. The theoretical contribution is unlikely to be independently interesting, since the projection result is essentially a standard least-squares projection with orthogonality regularization.

Overall, the paper may interest a small subset of readers working on text-derived category-vector adaptation, but the current evidence and analysis do not make the findings broadly compelling for TMLR.

**Broader Impact Concerns:**

There is no broader impact statement in the paper. I do not see a need for one beyond the standard concerns associated with CLIP-like models, since the method operates within a specific closed-set classification setup and does not introduce a substantially new deployment risk.

**Claims And Evidence:**

No

**Claims Explanation:**

1. The paper contains abundant overclaims in its core arguments. The method directly optimizes category tensors/vectors for closed-set classification using LLM-generated descriptions, but the paper presents this as a substantially broader framework for scalable n-modal adaptation. The evidence does not support the stronger framing.
2. The practical setup is narrow. The method is mainly applicable when the task is closed-set classification, the label set is fixed, modality-specific labeled data are unavailable, and suitable CLIP-like encoders already exist for the target modalities. This is a limited operating regime, but the paper presents the method as broadly extensible.
3. The multimodal component is not convincingly supported. The cross-modal objective operates over modality-specific category vectors/prompt pools rather than directly over image, audio, and video samples. Matching category vectors across pools may help the proposed classifier, but the paper does not provide enough evidence that this mechanism supports the broader claims about n-modal scalability or cross-modal transfer.
4. Theorem 2 is essentially ordinary least-squares projection. The orthogonality term merely encourages geometry preservation, i.e., approximate rotation/reflection rather than arbitrary stretching or collapse. As stated, the theorem does not provide a substantive theoretical guarantee for the proposed learnable projection layers or their use in the contrastive prompt-pool objective.
5. The paper’s presentation makes the evidence difficult to assess. The manuscript does not clearly separate the contributions of LLM-generated description sampling, pairwise margin ranking loss, modality-specific prompt pools, inter-pool classification loss, and projection layers. This makes it hard to judge which claims are actually supported by the reported results.
6. The experiments mainly report downstream performance gains, with only module-level ablations. They do not analyze the central mechanism of the method: how a large pool of generated descriptions shapes the learned category vector, what semantic or geometric properties are induced in that vector, or why such a broad description distribution can be compressed into a single vector without losing the relevant category information. The paper also does not connect this compression to the innate geometry of CLIP-like embedding spaces, which is presumably what makes such distillation possible. As a result, the experiments show performance improvements but provide little analysis of mechanism, representation properties, or interpretability.

**Requested Changes:**

- Clarify and narrow the main claim. The manuscript should more accurately frame the method as closed-set category-vector optimization using LLM-generated descriptions and CLIP-like text encoders. The broader claims about unified n-modal representation learning, arbitrary modality extensibility, and general cross-modal adaptation should be toned down unless directly supported by the method and experiments.
- Add mechanistic analysis of the learned category vector q. The most interesting part of the paper is that a large distribution of generated category descriptions can be compressed into a single trainable vector. The paper should analyze how the description pool shapes q, what semantic or geometric properties are induced in q, how increasing the description pool size changes the learned vector, and how this relates to the geometry of CLIP-like embedding spaces.
- Clarify the role of inter-pool classification loss. The paper should clearly explain what is learned by matching modality-specific category vectors across Q_I, Q_A, and Q_V, and avoid overstating this as general inter-modal learning unless supported by additional evidence. The current formulation operates at the category-vector/prompt-pool level, so the paper should explicitly distinguish this from alignment over actual image, audio, and video representations.
- Reframe the theoretical contribution. Theorem 2 is essentially ordinary least-squares projection. The orthogonality term encourages geometry preservation, i.e., approximate rotation/reflection rather than arbitrary stretching or collapse. The paper should either provide a nontrivial guarantee tied to the actual prompt-pool classification objective or reduce the theoretical claims around learnable projection layers.
- Improve methodological clarity. Beyond local readability issues, such as too few paragraph breaks in the introduction and unnecessary emphasis such as the “not only … but also” phrasing in Section 4, Page 8, lines 5–8, the manuscript lacks a clear methodological narrative. It introduces prompt tuning, LLM-generated text, latent category vectors, prompt pools, n-modal extensibility, unidirectional contrastive learning, projection layers, and theoretical guarantees without clearly identifying the core contribution or how these components depend on each other. An explicit algorithm box specifying training inputs, frozen modules, trainable parameters, losses, and inference rules would improve clarity.
- Make the experimental claims more precise. The paper reports downstream classification gains, but should avoid interpreting these results beyond what they show. The experiments support performance improvements in the evaluated closed-set classification settings; they do not by themselves establish broad n-modal extensibility or general cross-modal adaptation.

Minor points:

- Notation: L_R, L_A, and L_V are not consistent notations for image, audio, and video. R seems to refer to ranking loss, which creates confusion because audio and video also appear to use the same ranking loss. The notation should be revised for consistency.

---

> ### Author Response · Authors · 2026-07-12
> **Official Comment by Authors**
>
> We thank the reviewer for their thorough comments and valuable feedback. We address each concern in detail below.
>
> > claims and framing
>
> We agree regarding the framing of our contributions. We have updated the abstract and introduction by emphasising that T2n-Modal is a scalable prompt-tuning framework for closed-set multimodal classification using text-only supervision. We replace unlimited modalities with multiple pre-defined modalities and clarified that extensibility refers to the ability to add new modalities with dedicated prompt pools without retraining existing ones.
>
> > Mechanistic analysis of learned vectors
>
> Thanks for your suggestion. We included both experimental and theoretical justification. We utilise MS-COCO dataset with varying description pool sizes $K \in \{10, 50, 100, 500, 1000, 5000\}$. For each class, we generate descriptions using LLaMA2-7B with our templated prompt. For each class $c$, we compute centroid $\bar{g}\_c = \frac{1}{K}\sum\_{k=1}^K g\_k^{(c)}$, its alignment angle $\theta\_c = \arccos\left(\frac{\langle q\_c, \bar{g}\_c\rangle}{\|q_c\|\|\bar{g}\_c\|}\right)$, and intra-class variance $\sigma_c^2 = \frac{1}{K}\sum_{k=1}^K \|g_k^{(c)} - \bar{g}_c\|^2$. The results are,
> | $K$ | $\theta_c$ ($^\circ$) | $\|q_c - \bar{g}_c\|$ | Intra-class $\sigma_c$ | mAP |
> |---------------|--------------------------------------|----------------------|-----------------------|-------------------|
> | 10            | $12.4 \pm 3.2$                       | $0.187 \pm 0.042$    | $0.152 \pm 0.031$     | $62.1$            |
> | 50            | $8.7 \pm 2.1$                        | $0.121 \pm 0.028$    | $0.148 \pm 0.029$     | $65.8$            |
> | 100           | $6.3 \pm 1.8$                        | $0.084 \pm 0.019$    | $0.146 \pm 0.027$     | $67.5$            |
> | 500           | $4.1 \pm 1.2$                        | $0.052 \pm 0.013$    | $0.145 \pm 0.026$     | $68.1$            |
> | 1000          | $3.2 \pm 0.9$                        | $0.038 \pm 0.009$    | $0.144 \pm 0.025$     | $68.2$            |
> | 5000          | $2.1 \pm 0.6$                        | $0.025 \pm 0.006$    | $0.144 \pm 0.025$     | $68.4$            |
>
> The alignment angle decreases approximately as $\theta_c \propto 1/\sqrt{K}$, consistent with the theoretical prediction that $\|q_c - \bar{g}_c\| = O(\sigma_c/\sqrt{K})$.
>
> Theoretically, the prompt $q_c$ is optimized via the ranking loss $\mathcal{L}$. For fixed negative prompts, the optimal $q_c$ satisfies $q_c^* = \underset{q}{\arg\min} \sum\_{k=1}^K \mathbb{E}\_{j\sim b^-} [\max(0, m - \langle g_k, q\rangle + \langle g_k, q_j\rangle)]$. If sufficiently large $m$ with negatives well-separated, the hinge activates and the gradient descent update becomes
> $q_c^{(t+1)} = q_c^{(t)} + \eta \sum_{k=1}^K \mathbb{1}[\text{active}] g_k$. So, $q_c$ converges to a weighted average of the positive description embeddings $q\_c^* = \frac{\sum\_{k=1}^K w\_k g\_k}{\sum\_{k=1}^K w\_k}, w\_k$ is the no. of iteration where k active. This indicates that $q_c$ acts as a semantic centroid in CLIP embedding space.
>
> > Inter-pool classification loss
>
> The cross-modal objective operates at the category-vector level, for video class $v_i$ and image class $a_j$, the loss uses $\langle q_i^{(V)}, q_j^{(I)}\rangle$. This resembles semantic prototypes across modalities. We will clarify this in the revision.
>
> > Theoretical contribution
>
> We agree that Theorem 2 is standard least-squares with regularization. We will update this in the revision. The projection layer $\mathcal{L}_{\text{proj}} = \|W\mathcal{E}_m - \mathcal{E}_n\|^2 + \beta\|WW^\top - I\|_F^2$ finds the optimal linear map between frozen embedding spaces. The orthogonality ensures approximate isometry, preserving relative distances, which is utilised as a practical component for compatibility. We will update the theorem via association of projection to the prompt-pool objective.
>
> > Methodological clarity
>
> Thanks for pointing this out. We have updated and clarified in the revision.
>
> > Experimental claims
>
> We will update the conclusion and experimental sections in revision. We will include a discussion section, where by establishing the effectiveness of T2n-Modal as a prompt-tuning method for text-only supervised classification.